# Identification of neural progenitor cells and their progeny reveals long distance migration in the developing octopus brain

Astrid Deryckere[1†*], Ruth Styfhals[1,2], Ali Murat Elagoz[1], Gregory E Maes[3,4,5], Eve Seuntjens[1*]

[1]Laboratory of Developmental Neurobiology, Department of Biology, KU Leuven, Leuven, Belgium; [2]Department of Biology and Evolution of Marine Organisms, Stazione Zoologica Anton Dohrn, Naples, Italy; [3]Center for Human Genetics, Genomics Core, UZ-KU Leuven, Leuven, Belgium; [4]Centre for Sustainable Tropical Fisheries and Aquaculture, College of Science and Engineering, James Cook University, Townsville, Australia; [5]Laboratory of Biodiversity and Evolutionary Genomics, Department of Biology, KU Leuven, Leuven, Belgium

*For correspondence:
ad3846@columbia.edu (AD);
eve.seuntjens@kuleuven.be (ES)

Present address: †Department of Biological Sciences, Columbia University, New York, United States

Competing interests: The authors declare that no competing interests exist.

**Abstract** Cephalopods have evolved nervous systems that parallel the complexity of mammalian brains in terms of neuronal numbers and richness in behavioral output. How the cephalopod brain develops has only been described at the morphological level, and it remains unclear where the progenitor cells are located and what molecular factors drive neurogenesis. Using histological techniques, we located dividing cells, neural progenitors and postmitotic neurons in *Octopus vulgaris* embryos. Our results indicate that an important pool of progenitors, expressing the conserved bHLH transcription factors *achaete-scute* or *neurogenin*, is located outside the central brain cords in the lateral lips adjacent to the eyes, suggesting that newly formed neurons migrate into the cords. Lineage-tracing experiments then showed that progenitors, depending on their location in the lateral lips, generate neurons for the different lobes, similar to the squid *Doryteuthis pealeii*. The finding that octopus newborn neurons migrate over long distances is reminiscent of vertebrate neurogenesis and suggests it might be a fundamental strategy for large brain development.

## Introduction

Cephalopod mollusks represent an invertebrate lineage that exhibits morphological as well as behavioral complexity reminiscent of vertebrates. Studying species from this group thus brings an opportunity to understand the genetic drivers of the development of nervous systems that evolved convergently with vertebrates. The adult cephalopod mollusk *Octopus vulgaris* has a highly centralized brain containing about 200 million nerve cells in the supra- and subesophageal mass and two optic lobes (*Young, 1963*; *Young, 1971*), yet the cellular and molecular mechanisms driving brain development remain poorly understood. At hatching, the *O. vulgaris* brain counts about 200,000 cells and occupies roughly one fourth of the total body, indicating extensive embryonic neurogenesis (*Budelmann, 1995*; *Giuditta et al., 1971*; *Packard and Albergoni, 1970*). In general, neural progenitor cells are generated from ectodermal cells and divide symmetrically and asymmetrically to generate all neurons of the nervous system (*Florio and Huttner, 2014*; *Kriegstein and Alvarez-Buylla, 2009*; *Wodarz and Huttner, 2003*). In clades harboring species with diffuse nerve nets such as cnidaria and hemichordates, the proliferating neural progenitor cells are distributed throughout the ectoderm generating local neurons, while in (sub)phyla with a centralized nervous system including vertebrates, arthropods and some annelids, the neural progenitor cells are grouped in the

**eLife digest** Octopuses have evolved incredibly large and complex nervous systems that allow them to perform impressive behaviors, like plan ahead, navigate and solve puzzles. The nervous system of the common octopus (also known as *Octopus vulgaris*) contains over half a billion nerves cells called neurons, similar to the number found in small primates. Two thirds of these cells reside in the octopuses' arms, while the rest make-up a central brain that sits between their eyes.

Very little is known about how this central brain forms in the embryo, including where the cells originate and which molecular factors drive their maturation in to adult cells. To help answer these questions, Deryckere et al. studied the brain of *Octopus vulgaris* at different stages of early development using various cell staining and imaging techniques. The experiments identified an important pool of dividing cells which sit in an area outside the central brain called the 'lateral lips'. In these cells, genes known to play a role in neural development in other animals are active, indicating that the cells had not reached their final, mature state.

In contrast, the central brain did not seem to contain any of these immature cells at the point when it was growing the most. To investigate this further, Deryckere et al. used fluorescent markers to track the progeny of the dividing cells during development. This revealed that cells in the lateral lips take on a specific neuronal fate before migrating to their target region in the central brain. Newly matured neurons have also been shown to travel large distances in the embryos of vertebrates, suggesting that this mechanism may be a common strategy for building large, complex brains.

Although the nervous system of the common octopus is comparable to mammals, they evolved from a very distant branch of the tree of life; indeed, their last common ancestor was a worm-like animal that lived about 600 million years ago. Studying the brain of the common octopus, as done here, could therefore provide new insights into how complex nervous systems, including our own, evolved over time.

neurectoderm. The proliferating neural progenitor cells either remain on the apical surface of the neurectoderm in vertebrates and annelids, or internalize as is the case for insects (*Cunningham and Casey, 2014*; *Götz and Huttner, 2005*; *Hartenstein and Stollewerk, 2015*; *Lowe et al., 2003*; *Meyer and Seaver, 2009*; *Rentzsch et al., 2017*; *Simionato et al., 2008*; *Taverna et al., 2014*; *Urbach and Technau, 2004*). In addition, long-distance migration of neurons has been described for developing vertebrate brains, in which neurons born in different zones follow long trajectories to their final location, where they intermingle to form complex circuits (*García-Moreno and Molnár, 2020*). Although neuronal migration has been described in developing invertebrate nervous systems as well, this process is generally limited to restricted cell populations, for example the Q, CAN and HSN neuroblasts in *Caenorhabditis elegans* (*Blelloch et al., 1999*; *Forrester et al., 1998*; *Montell, 1999*), or to short-range migratory events, for example in the *Drosophila* visual system (*Apitz and Salecker, 2015*; *Bhat, 2007*; *Morante et al., 2011*).

Molecular studies in vertebrates, *Drosophila*, *C. elegans*, but also cnidarians, have revealed a set of regulatory transcription factors involved in neurogenesis (*Arendt et al., 2008*; *Bertrand et al., 2002*; *Galliot et al., 2009*; *Hirth, 2010*; *Layden et al., 2012*). First, group B SRY-related HMG box genes (*soxB* genes) regulate the generation of the neurectoderm and also maintain neurectodermal cells in an undifferentiated and proliferative state (*Guth and Wegner, 2008*; *Sarkar and Hochedlinger, 2013*). After neurectoderm formation, members of the superfamily of basic helix-loop-helix (bHLH) transcription factors control the specification of neural progenitor cells and also activate neuronal differentiation pathways (*Bertrand et al., 2002*; *Vervoort and Ledent, 2001*). The bHLH protein superfamily is divided in groups A-F, based on structural and biochemical properties. bHLH genes such as *atonal*, *neurogenic differentiation* (*neuroD*) and *neurogenin* (*ngn*), and *achaete-scute* (*asc*) complex members are classified in group A. The role of bHLH genes is best described in nervous system development where members of this group A (and some of group B) regulate the patterning, differentiation and specification of neurons, from sponges to primates (*Powell and Jarman, 2008*; *Simionato et al., 2007*; *Vervoort and Ledent, 2001*). Although the level of differentiation of progenitor cells in which these factors are expressed and the sequential expression of bHLH genes

differs slightly across species, they all steer progenitor cells towards a neural fate. After this neural progenitor commitment, regulatory genes such as *musashi*, *prospero*, and *embryonic lethal abnormal vision* (*elav*) (vertebrate *hu*), will activate programs to initiate differentiation of progenitor cells into neurons (*Choksi et al., 2006*; *Okano et al., 2002*; *Pascale et al., 2008*). In the end, postmitotic neurons will mature, form synapses, and produce neurotransmitters to form a fully functional central nervous system (CNS).

These neurogenic processes are poorly studied in lophotrochozoans, and above all, in lophotrochozoans with a complex nervous system such as *O. vulgaris*. How and where neurons are generated, whether they migrate and what factors drive their differentiation and integration remain largely unknown. One lineage tracing study in the squid *Doryteuthis pealeii* revealed that cells surrounding the eye placode contribute to the anterior chamber organ, the supraesophageal mass, the buccal mass and the buccal ganglion (*Koenig et al., 2016*). Our study is the first combining gene expression data with functional lineage tracing to explain neurogenesis in cephalopods. The high fecundity of *O. vulgaris*, spawning small and transparent eggs that can be kept in a remote tank system, as well as the available genomic data recently made it a tractable species for embryonic studies (*Deryckere et al., 2020*; *Zarrella et al., 2019*). Here, we have visualized the anatomy of brain development in 3D using light sheet microscopy. Expression analysis of several transcription factors, proliferation and neural markers revealed the spatial and temporal pattern of *O. vulgaris* brain development from stage IX onwards. Lastly, lineage-tracing studies using CFDA-SE flash labeling proved that the neurogenic area is spatially and temporally patterned. Our data suggest that *O. vulgaris* deploys a neurogenic transcription factor sequence during neurogenesis, as well as extensive neural migration, that are reminiscent of vertebrate mechanisms of large brain generation.

## Results

### The rapidly growing octopus brain suggests massive embryonic neurogenesis

*O. vulgaris* displays a direct embryonic development, giving rise to actively feeding paralarvae. Their embryonic development takes approximately 40 days at 19℃ and has been classified in 20 major stages I-XX, with some stages subdivided in an early and late part (*Deryckere et al., 2020*; *Naef, 1928*). Organogenesis starts at Stage VII.2 and can be split in early, mid-, and late phases (*Figure 1A*). We used paraffin sectioning and light sheet imaging combined with DAPI staining to document the different steps of brain development in 2 and 3 dimensions. In the early organogenesis phase, the brain anlagen (cordal) are elongated and interconnected (in three dimensions) (*Figure 1A,B*) as also described by Shigeno et al. for *Octopus bimaculoides* (*Shigeno et al., 2015*). On the most anterior side, lateral and posterior to the mouth and foregut, the cerebral cord (CC) will give rise to the lobes of the supraesophageal mass (SEM), while on the posterior side of the embryo, the pallioviscerial (PVC) and pedal (PC) cords will form the subesophageal mass (SUB). The two bilateral optic cords (OC) will differentiate and grow to form the optic lobes (OL) (*Figure 1A,C*). At hatching, the central brain that now surrounds the esophagus is considerably big and contains densely packed nuclei (*Figure 1D*). In contrast to the growing brain cords, the tissue adjacent to the eye primordia first grows in size from Stage VII.2 to Stage XV.2, and then shrinks to disappear at hatching, suggesting this tissue might contribute to the inner head structures (*Figure 1A*). This tissue was first described as 'Kopflappen' by Marquis and later as 'anterior chamber organ' by *Koenig et al., 2016*; *Marquis, 1989*; *Shigeno et al., 2001*; *Yamamoto et al., 2003*. However, the term 'anterior chamber organ' was introduced by Young, referring to neurovenous tissue adjacent to the adult eye, that presumably regulates the fluid content between the lens and cornea (*Young, 1970*). The term was most likely wrongly adopted embryonically as the tissue at that stage is not restricted to the anterior side of each eye, but is rather placed lateral to the eye fields. Based on 3D light sheet imaging, we could clearly show that the structure is also connected to the central brain, through a stream-like transition zone. Because of its position and shape, we renamed the tissue to 'lateral lips' (LL, indicated in mint green in *Figure 1C,E*) and introduced the terms 'anterior transition zone' (ATZ) and 'posterior transition zone' (PTZ) for the region that interconnects the lateral lips with the central brain on the anterior or posterior side of the embryo, respectively (*Figure 1C,F*).

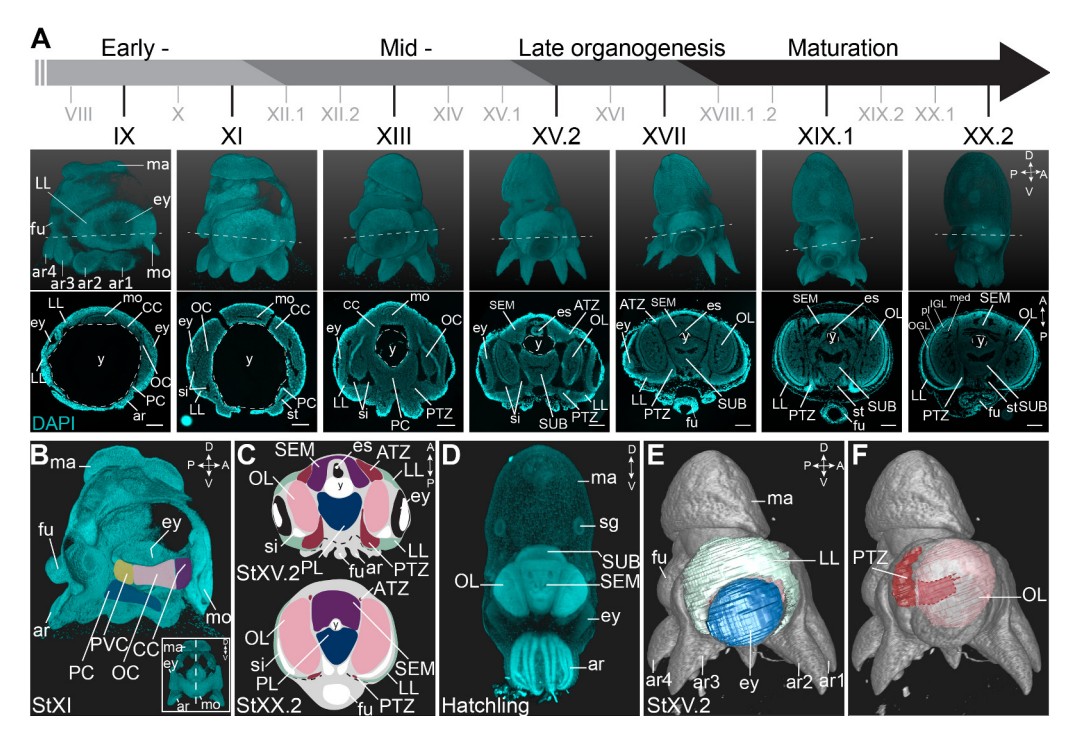

**Figure 1.** Overview of the developing *O. vulgaris* embryo and its nervous system. (**A**) Overview of *O. vulgaris* embryonic development from Stage IX to Stage XX.2, covering early-, mid-, and late- organogenesis events and maturation. Surface renderings after DAPI staining are shown in the upper panels (lateral view, dorsal side up) and representative transversal sections in the lower panels (anterior side up). The OCs that give rise to the OLs develop medially from the eye primordia. The CC that generates the SEM develops next to the external mouth and the PC and PVC that give rise to the SUB develop on the posterior side. The dashed lines on the surface renderings indicate the sectioning plane of the transversal sections. Scale bars represent 100 μm. (**B**) Surface rendering after DAPI staining at Stage XI, showing that the central brain cords are connected and encircle the yolk. Prospective cords are pseudo-colored: CC in purple, OC in pink, PVC in yellow and PC in blue. (**C**) Schematic of the octopus head region late-organogenesis and at hatching (ATZ and PTZ in red, LL in mint green, OL in pink, SEM in purple, PL of the SUB in blue). (**D**) Maximum projection after DAPI staining of a hatchling showing the densely nucleated central brain from the anterior side. (**E-F**) 3D reconstruction of the eye (blue), LL (mint green), OL (pink), and the PTZ (red) in a Stage XV.2 embryo (DAPI in gray). A, anterior; ar, arm; ATZ, anterior transition zone; CC, cerebral cord; D, dorsal; es, esophagus; ey, eye; fu, funnel; IGL, inner granular layer; LL, lateral lips; ma, mantle; med, medulla; mo, mouth; OC, optic cord; OGL, outer granular layer; OL, optic lobe; P, posterior; PC, pedal cord; pl, plexiform layer; PL, pedal lobe; PTZ, posterior transition zone; PVC, pallliovisceral cord; PVL, palliovisceral lobe; SEM, supraesophageal mass; sg, stellate ganglion; si, sinus ophthalmicus; st, statocyst; SUB, subesophageal mass; V, ventral; y, yolk.

## The developing octopus brain shows signs of early neuronal differentiation

In order to map the (early) patterns of neuronal development in *O. vulgaris*, we studied the expression of the pan-neuronal *elav* gene. ELAV/Hu RNA-binding proteins are a family of splicing factors, predominantly present in differentiating neurons, from the moment they exit the cell cycle (*Colombrita et al., 2013*). Through blast searches against the full-length transcriptome of *O. vulgaris* embryos and paralarval brains generated in this study (see Materials and methods), we identified three candidate *elav* transcripts. We performed a phylogenetic analysis on these ELAV proteins together with 27 ELAV sequences from 20 other species, five non-neural ELAV sequences from four other species and five PolyA-binding sequences in order to root the tree (*Figure 2—figure supplement 1*). One of the *O. vulgaris* ELAV candidate proteins nested with vertebrate HuC/D and invertebrate neural ELAV homologs. We refer to this sequence as *Ov*-ELAV. The two other candidate proteins could be identified as non-neural (*Figure 2—figure supplement 1*). We also performed a conserved domain (CD)-Search and detected an ELAV/HuD family splicing factor domain which is characteristic of ELAV proteins.

We mapped the expression of *Ov-elav* using in situ hybridization at embryonic stages IX, XI, XIII, XV.2, XVII, XIX.1, and XX.2 (*Figure 2A–G*, *Figure 2—figure supplement 2*). These stages serve as a

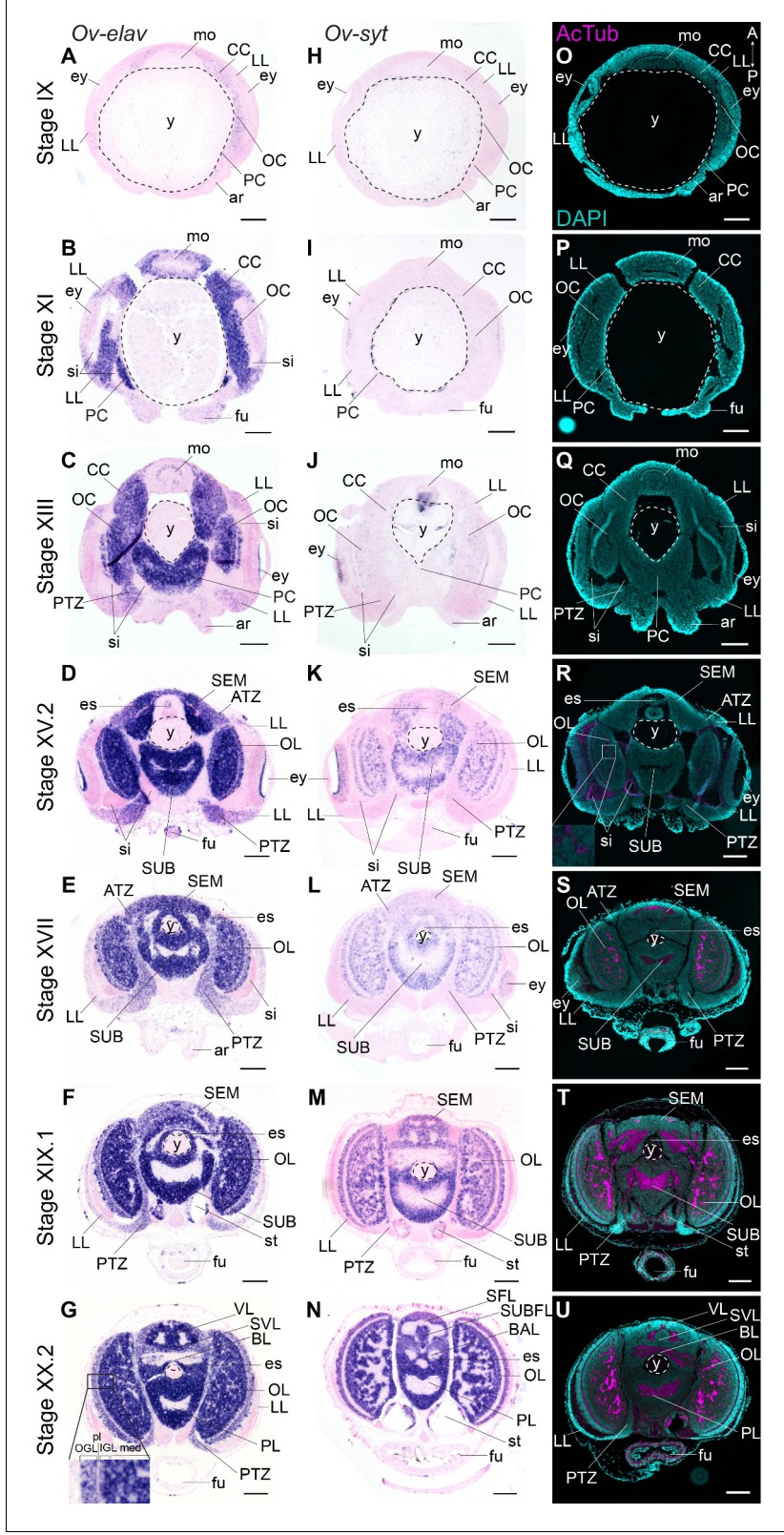

**Figure 2.** Formation of the *O. vulgaris* brain. In situ hybridization of *Ov-elav* (left column) and *Ov-syt* (middle column) and immunoreactivity against acetylated alpha tubulin (right column) on transversal sections of embryos at Stage IX, XI, XIII, XV.2, XVII, XIX.1, and XX.2 with anterior up and posterior down. (**A-G**) *Ov-elav* expression levels are clearly elevated from Stage XI onwards and are generally highest in the developing brain cords and central

*Figure 2 continued on next page*

*Figure 2 continued*

brain masses, intermediate in the transition zones and low in the lateral lips. (**H-N**) *Ov-syt* can be detected first in the retina at Stage XI and in the outer layers of the optic cord at Stage XIII. From Stage XV.2 onwards, transcripts are present in all brain masses. (**O-U**) Acetylated alpha-tubulin is present in the optic lobes from Stage XV.2 onwards (low level, magnified box) and in the supra- and subesophageal masses from Stage XVII onwards. See **Figure 2—figure supplement 1** for phylogenetic reconstruction of *Ov-elav* and **Figure 2—figure supplements 2** and **3** for complementary panels of *Ov-elav* and *Ov-syt* expression. Scale bars represent 100 µm. A, anterior; ar, arm; ATZ, anterior transition zone; BAL, buccal lobe; BL, basal lobe; CC, cerebral cord; es, esophagus; ey, eye; fu, funnel; IFL, inferior frontal lobe; igl, inner granular layer; LL, lateral lips; med, medulla; mo, mouth; OC, optic cord; ogl, outer granular layer; OL, optic lobe; P, posterior; PC, pedal cord; pl, plexiform layer; PL, pedal lobe; PTZ, posterior transition zone; SEM, supraesophageal mass; SFL, superior frontal lobe; sg, stellate ganglion; si, sinus ophthalmicus; st, statocyst; SUB, subesophageal mass; SUBFL, subfrontal lobe; SVL, subvertical lobe; VL, vertical lobe; y, yolk.

The online version of this article includes the following figure supplement(s) for figure 2:

**Figure supplement 1.** Phylogeny *Ov*-ELAV.

**Figure supplement 2.** Complementary panels to *Figure 2* of *Ov-elav* expression in the head region of developing *O. vulgaris* embryos.

**Figure supplement 3.** Complementary panels to *Figure 2* of *Ov-syt* expression in the head region of developing *O. vulgaris* embryos.

---

starting point for the characterization of nervous system development during organogenesis (Stage IX-XVII) and maturation phases (Stage XIX.1 and XX.2) of embryonic development. *Ov-elav* transcripts were detected throughout development with little expression at Stage IX in the cerebral, pallovisceral, pedal, and optic cords (*Figure 2A*). Strong staining was observed from Stage XI onwards, with low-level staining in the lateral lips surrounding the eye placode and more intense staining in all cords and surrounding the mouth (*Figure 2B*). A similar pattern was observed at subsequent stages with highest staining intensity in the central brain cords/masses and low intensity staining in the lateral lips. At Stage XIII, a region in between the lateral lips and the central brain on the posterior side of the embryo showed intermediate *Ov-elav* expression. This region corresponds to the posterior transition zone (PTZ), introduced earlier (*Figure 2C*). From Stage XV.2 onwards, cells on the anterior side of the embryo and laterally from the supraesophageal mass, corresponding to the anterior transition zone (ATZ), showed intermediate *Ov-elav* expression levels as well (*Figure 2D,E*). In the optic lobes at stages XIX.1 and XX.2, *Ov-elav* expression was highest in the medulla and inner granular layer, but lower in the outer granular layer. The ATZ and PTZ shrink toward the end of embryonic development and seem to become integrated in the developing brain (*Figure 2F,G*). In summary, our data indicate that neuronal differentiation is present at early organogenesis phases already, and the transition zones as well as the growing cords and masses contain large numbers of differentiating neurons.

To address the maturation of these neurons, we investigated the expression pattern of *synaptotagmin*. Synaptotagmin is a presynaptic calcium sensor necessary for neurotransmitter release, present in mature synapses and thus in differentiated, functional neurons (*Poskanzer et al., 2003*). In addition, we also visualized neuronal somata and neuropil using immunoreactivity against acetylated alpha-tubulin, as previously described for multiple cephalopod species (*Figure 2H–U, Figure 2—figure supplement 3; Jung et al., 2018; Kingston et al., 2015; Scaros et al., 2018; Shigeno et al., 2015; Shigeno and Yamamoto, 2002; Wollesen et al., 2009; Wollesen et al., 2012*). In *O. vulgaris*, we observed expression of *Ov-syt* in the retina from Stage XI onwards (*Figure 2H,I*). In the central brain, it was first expressed in the outer layers of the optic cord at Stage XIII, and in the optic lobe medulla, supra- and subesophageal masses from Stage XV.2 onwards, pointing to a sequential maturation (*Figure 2J–N*). First low-level immunoreactivity against acetylated alpha tubulin was found from Stage XV.2 onwards within the optic lobes (*Figure 2O–U*). We therefore used the adult terminology (optic lobe, supra- and subesophageal mass) for CNS annotation from this stage onwards, since the embryonic cords demonstrate clear signs of differentiation.

## The lateral lips harbor proliferating cells, whereas the developing cords are mostly postmitotic

The developing octopus brain thus consists mainly of *Ov-elav* expressing cells, and it remains unclear where these cells are initially generated. A previous report in the squid *D. pealeii* suggested that cells surrounding the eye placode contribute to the brain (*Koenig et al., 2016*). This area might therefore harbor a proliferative zone contributing to embryonic neurogenesis. In order to locate proliferating cells that might contribute to CNS development, we used immunoreactivity against phospho-histone H3 (PH3) and mapped the expression pattern of *Ov-pcna*.

PCNA is a nuclear protein required for DNA replication and repair, and functions as a cofactor of DNA polymerase-delta in eukaryotes and archaea (*Barry and Bell, 2006*; *Baserga, 1991*; *Prelich et al., 1987*). Expression of *pcna* is tightly regulated and peaks at late $G_1$ and S phases of the cell cycle (*Santos et al., 2015*). The expression of the *O. vulgaris* homolog was mapped using in situ hybridization at embryonic Stages VII.2, IX, XI, XIII, XV.2, XVII, XIX.1, and XX.2. *Ov-pcna* transcripts were detected throughout embryonic development (*Figure 3A–D,I–L*). While at Stages VII.2 and IX, transcripts were found in most embryonic tissues (mouth apparatus, retina, lateral lips and cords, in the latter at lower intensity), transcripts gradually disappeared in the optic, cerebral, palliovisceral and pedal cords from Stage XI onwards (*Figure 3A–C*). At Stage XI, most *Ov-pcna* expressing cells that were found adjacent to the eye, located to an area that is *Ov-elav* negative and that might be part of the epithelium lining the sinus ophthalmicus (*Figure 3C*). *Ov-pcna* transcripts were dispersed throughout the lateral lips and were absent from the transition zones and central brain at Stages XIII-XX.2 (*Figure 3D,I–L*). Since transcripts of *Ov-pcna* might still be present at low level in cells that finished DNA replication, we also mapped the phosphorylation of histone H3 during embryonic development (*Figure 3E–H,M–P*). Histone H3 is phosphorylated on serine 10 and serine 28 during early mitosis and dephosphorylated at the end of mitosis in eukaryotes (*Hans and Dimitrov, 2001*; *Prigent and Dimitrov, 2003*; *Wei et al., 1998*). Immunohistochemistry on embryonic tissue from Stage VII.2 to XX.2 showed that mitotic activity in the embryonic head region was high at the beginning of organogenesis, with the highest level of dividing cells present in the mouth apparatus, at the apical side in the developing retina and spread throughout the lateral lips (*Figure 3E–H,M–P*). Consistent with *Ov-pcna* expression, the number of PH3-positive cells in the cerebral, optic, palliovisceral and pedal cords was limited, with possibly very few, single, dim and seemingly randomly organized positive cells present until Stage XIII (*Figure 3E–H*). At late organogenesis and maturation stages, PH3-positive cells located to the lateral lips and were absent from the central brain (*Figure 3M–P*). Given that the *elav*-positive areas are almost devoid of dividing cells, and the transition zone represents an area connecting the mitotically active lateral lips to the growing CNS, our data strongly suggest that the lateral lips represent a major octopus neurogenic zone.

## Conserved sequence of neurogenic transcription factor gene expression during *O. vulgaris* brain development

In order to find molecular support for the neurogenic character of the lateral lip cells, we mapped the expression of conserved genes involved in neural stem cell specification and differentiation, namely *soxB1*, *ascl1*, *ngn*, and *neuroD*.

First, to spatially map the extent of the neurectoderm, we studied the expression of *soxB1*. *Sox* genes have been divided into seven groups (A-G) based on the sequence of their high mobility group (HMG)-box that binds to DNA in a sequence-specific manner (*Sasai, 2001*; *Wegner, 1999*). Members of the SoxB group play important roles in (early) neurogenesis in various bilaterians and can be further divided in two sub-groups SoxB1 and SoxB2 based on sequence differences outside the HMG-box and on their activity (*Bowles et al., 2000*; *She and Yang, 2015*). Using BLASTn and tBLASTn searches, we identified *O. vulgaris* members for groups B-E, but not for group A which is mammalian specific or group G which is also restricted to particular vertebrate lineages (*Bowles et al., 2000*; *Heenan et al., 2016*). We could also not identify SOXF family members in the available cephalopod transcriptomes of *O. vulgaris*, *O. bimaculoides*, *Sepia officinalis*, and *Euprymna scolopes* (*Figure 4—figure supplement 1*). Based on phylogenetic analysis including 31 SOXB, 3 SOXC, 4 SOXD, 4 SOXE, and 3 SOXF amino acid sequences from other species together with 5 TCF/LEF sequences to root the tree, we identified single *O. vulgaris* homologs for SOXC, D and E

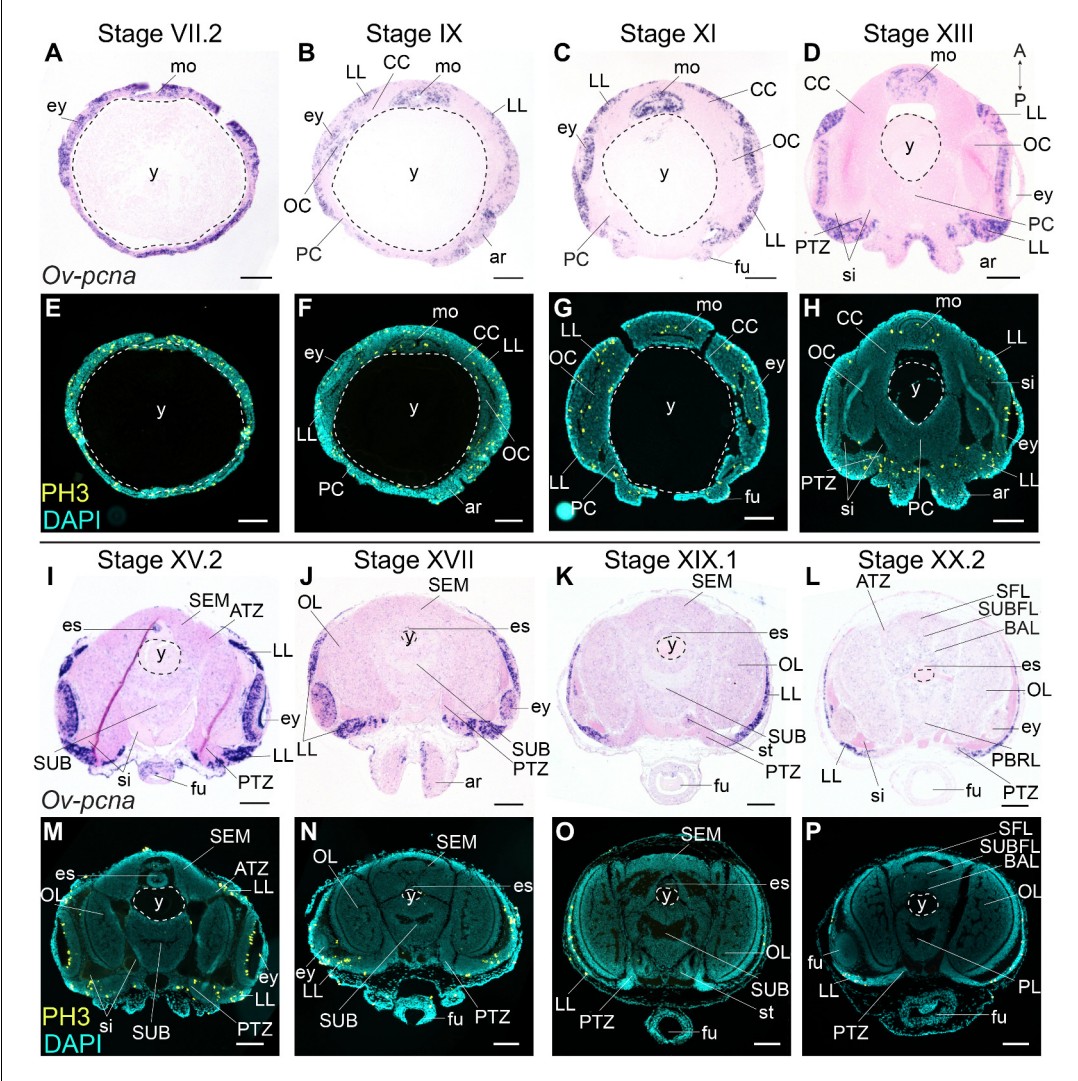

**Figure 3.** Cell division in the developing *O. vulgaris* embryo. Expression of *Ov-pcna* (upper panels) and immunoreactivity against PH3 (lower panels) from Stage VII.2 to Stage XX.2. (A-D, I-L) *Ov-pcna* expression is broad at Stages VII.2 and IX and gets restricted to the lateral lips, mouth region and retina at subsequent stages. (E-H, M-P) Similarly, PH3-positive cells are abundant in the lateral lips, in the mouth region and on apical side in the retina. Very few cells in the developing central brain are PH3 positive or express *Ov-pcna*. Scale bars represent 100 µm. Abbreviations as in *Figure 2*; PBRL, prebrachial lobe.

and two SOXB homologs (*Figure 4—figure supplement 1*). One *O. vulgaris* SOXB protein nested within the SOXB2 clade and the other one within the SOXB1 clade, the latter named *Ov*-SOXB1. CD-Search of this *Ov*-SOXB1 protein revealed the highly conserved SOX-TCF HMG-box and SOXp superfamily domains.

*Ov-soxB1* transcript levels were high in the head region at all embryonic stages examined (*Figure 4—figure supplement 2*). At Stage IX, *Ov-soxB1* was expressed in the lateral lips, eyes and surface ectoderm surrounding the mouth apparatus (esophagus, radula, salivary gland). In the developing central brain, *Ov-soxB1* was expressed in the cerebral cord, but not in the palliovisceral, pedal and optic cords (*Figure 4—figure supplement 2A–D*). At Stage XI, *Ov-soxB1* was still highly expressed in the lateral lips and cerebral cord, and transcripts were now also present in the optic cord in a patched pattern (*Figure 4—figure supplement 2E–H*). At Stage XIII, *Ov-soxB1* transcripts were most highly expressed in the cerebral cord and showed a similar patched pattern in the optic cords. Transcripts were also present in the pedal cord and limited in the palliovisceral cord (*Figure 4—figure supplement 2I–L*). At Stage XV.2, *Ov-soxB1* expression was high in the lateral lips

and supraesophageal mass. In the subesophageal mass, *Ov-soxB1*-positive cells were more spread and in the optic lobes, *Ov-soxB1* transcripts showed the highest expression in the inner granular layer, while expression was more distributed in the outer granular layer and medulla (*Figure 4—figure supplement 2M–P*). At Stage XVII, *Ov-soxB1* transcripts were still numerous in the central brain, which has started to form neuropil (*Figure 4—figure supplement 2Q–T*). At Stages XIX.1 and XX.2, *Ov-soxB1* was expressed in the lateral lips, the supra- and subesophageal mass and optic lobes where it was most highly expressed in the inner granular layer (*Figure 4—figure supplement 2U-b*). Taken together, *Ov-soxB1* was not only expressed in putative stem cells, but also in postmitotic (*Ov-elav+*) areas, as has been described for other invertebrate species.

To identify a transcription factor that would mark neural progenitors only, we mapped the expression of two proneuronal bHLH group A transcription factors *ascl1* and *neurogenin*, and one neuronal differentiation bHLH transcription factor *neuroD*. By sequence similarity searches, we identified *O. vulgaris* homologs for *achaete-scute*, *neurogenin* and *neuroD*. Phylogenetic analysis of the atonal-related bHLH transcription factors including 15 NEUROD and 8 NEUROGENIN/TAP protein sequences from other species effectively assigned *Ov-neuroD* and *Ov-ngn* to their respective subfamilies (*Figure 4—figure supplement 3*). bHLH proteins possess a bHLH domain which is involved in DNA binding and dimerization (*Jones, 2004*; *Murre et al., 1989*). CD-Search identified such conserved domains in both sequences. In addition, phylogenetic analysis for the achaete-scute-related bHLH gene *Ov-ascl1* together with 27 other ASCa protein sequences and 10 ASCb protein sequences placed *Ov*-ASCL1 within the proneuronal ASCa subgroup of the bHLH superfamily (*Figure 4—figure supplement 4*). CD-Search revealed a highly conserved DNA-binding bHLH domain.

Achaete-scute homologs are involved in vertebrate neural identity determination and are key regulators in *Drosophila* neuroblast generation (*Cabrera et al., 1987*; *Skeath and Carroll, 1992*; *Vervoort and Ledent, 2001*). In *O. vulgaris*, *Ov-ascl1* transcripts were detected throughout embryonic development at all stages tested (*Figure 4A–G*, *Figure 4—figure supplement 5*). At Stage IX, *Ov-ascl1* was expressed in the lateral lips surrounding the eye primordia. Expression could also be observed in retinal cells of the developing eyes and in the tissue delineating the mouth apparatus, and was absent from the cerebral, pedal, palliovisceral and optic cords (*Figure 4A*). At subsequent stages of organogenesis (XI, XIII, XV.2, and XVII), *Ov-ascl1* was expressed in the lateral lips and was absent from the developing cords/brain lobes (*Figure 4B–E*, low level staining that did not consistently appear in all replicates was considered background). During maturation and right before hatching, the number of cells expressing *Ov-ascl1* significantly decreased with the thinning lateral lips and transcripts were still absent from the optic lobes, supra- and subesophageal masses (*Figure 4F,G*). Similar to *achaete-scute*, homologs of *neurogenin* are highly conserved proneural genes expressed in early nervous system development, before neuronal differentiation in vertebrates, but also annelids (*Simionato et al., 2007*; *Simionato et al., 2008*; *Sur et al., 2017*; *Vervoort and Ledent, 2001*). In *O. vulgaris* embryos, we found *Ov-ngn* transcripts in the lateral lips throughout organogenesis and maturation phases, while transcripts were absent from the developing brain cords (Stage IX-XX.2, *Figure 4H–N*, *Figure 4—figure supplement 6*, low level staining that did not consistently appear in all replicates was considered background). In addition, very few *Ov-ngn* transcripts were present in both transition zones at Stages XIII to XX.2 (*Figure 4J–N*). Equivalent to *Ov-ascl1*, the number of *Ov-ngn* expressing cells decreased considerably in the maturation phase with the thinning of the lateral lips and at Stage XX.2, only few cells were expressing *Ov-ngn* (*Figure 4M,N*). In summary, expression of both *Ov-ascl1* and *Ov-ngn* consistently marks the proliferative lateral lips, with *Ov-ascl1* expressing cells being more numerous compared to *Ov-ngn* expressing cells.

The second atonal-related bHLH transcription factor studied here was NeuroD (*Figure 4O–U*, *Figure 4—figure supplement 7*). Its vertebrate and annelid homologs are expressed at the time neurons differentiate (*Sur et al., 2017*; *Vervoort and Ledent, 2001*). At the beginning of organogenesis (Stage IX and XI), *Ov-neuroD* was expressed in the cerebral, palliovisceral, pedal and optic cords of the central nervous system. Transcripts were also detected in the lateral lips, although in fewer cells compared to the cords (*Figure 4O,P*). At Stage XI in the optic cords, a medio-lateral expression gradient was observed with high *Ov-neuroD* at the medial side and low *Ov-neuroD* at the lateral side, closer to the eyes. *Ov-neuroD* also seemed more highly expressed in the cerebral and palliovisceral cords compared to the pedal cord (*Figure 4P*). At Stage XIII, *Ov-neuroD* transcripts were absent from the region of *Ov-ascl1* expressing cells in the lateral lips, but were present

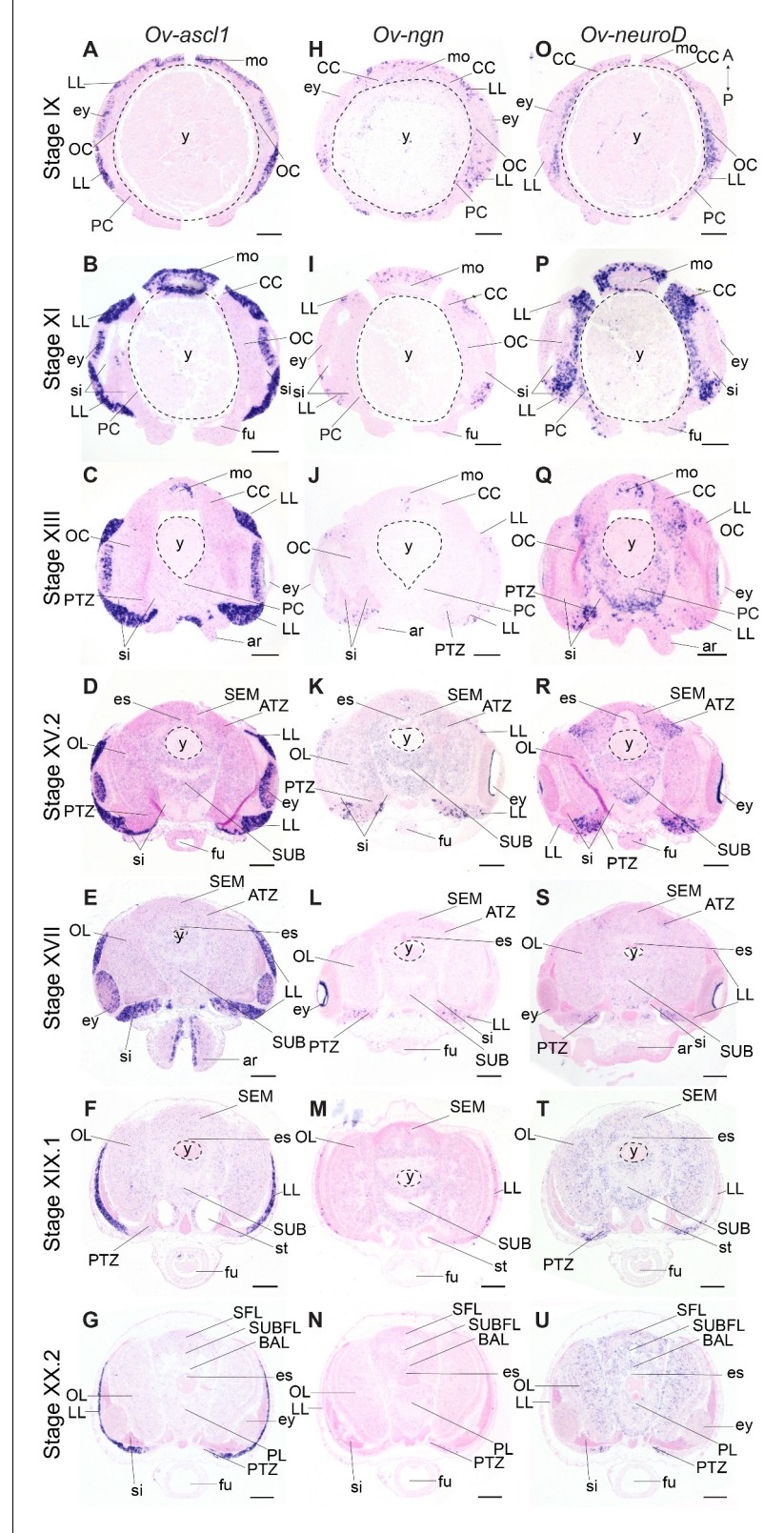

**Figure 4.** Expression of *O. vulgaris* proneuronal and neuronal differentiation bHLH genes during embryonic development. Expression of *Ov-ascl1*, *Ov-ngn*, and *Ov-neuroD* on paraffin sections from Stage IX to Stage XX.2. (A-G) *Ov-ascl1* is highly expressed in the lateral lips and retina at all stages. Overall, the number of *Ov-ascl1* positive cells increases during organogenesis, reaching a peak at Stage XV.2. (H-N) Expression of *Ov-ngn* is

*Figure 4 continued*

restricted to cells in the lateral lips and a limited number of cells in the transition zones at all stages. (**O-U**) *Ov-neuroD* is expressed at low level in the lateral lips at Stage IX and XI, but not in its most outer cell layers. At subsequent stages, *Ov-neuroD* is highly expressed in the transition zones. In the central brain, *Ov-neuroD* transcripts are present in the cerebral, optic, and palliovisceral (**Figure 4—figure supplement 5**) cords at Stage IX and also in the pedal cord from Stage XI onwards. Expression in the cords decreases over the course of development. See **Figure 4—figure supplements 1** and **2** for phylogenetic reconstruction and expression pattern of *Ov-soxB1*, **Figure 4—figure supplements 3** and **4** for phylogenetic reconstruction of *Ov-ngn*, *Ov-neuroD*, and *Ov-ascl1* and **Figure 4—figure supplements 5**, **6** and **7** for complementary panels of *Ov-ascl1*, *Ov-ngn*, and *Ov-neuroD* expression. Scale bars represent 100 µm. Abbreviations as in **Figure 2**.

The online version of this article includes the following figure supplement(s) for figure 4:

**Figure supplement 1.** Phylogeny *Ov*-SOXB1.
**Figure supplement 2.** Expression of *Ov-soxB1* during *O. vulgaris* embryonic development.
**Figure supplement 3.** Phylogeny of *Ov*-NEUROD and *Ov*-NGN.
**Figure supplement 4.** Phylogeny of *Ov*-ASCL1.
**Figure supplement 5.** Complementary panels to **Figure 4** of *Ov-ascl1* expression in the head region of developing *O. vulgaris* embryos.
**Figure supplement 6.** Complementary panels to **Figure 4** of *Ov-ngn* expression in the head region of developing *O. vulgaris* embryos.
**Figure supplement 7.** Complementary panels to **Figure 4** of *Ov-neuroD* expression in the head region of developing *O. vulgaris* embryos.

adjacent to those cells on the posterior side of the embryo next to the sinus ophthalmicus, and mark the posterior transition zone. In addition, *Ov-neuroD* expression in the cerebral, palliovisceral and optic cords was significantly reduced compared to earlier stages and expression in the pedal cord was elevated (**Figure 4Q**). At Stages XV.2 and XVII, high level *Ov-neuroD* expression marked both the anterior and posterior transition zones. In the central brain, transcripts were present at low level in the optic lobes on the medial side, and in the outer layers of the supra- and subesophageal masses (**Figure 4R,S**, low level staining in the developing cords/lobes that did not consistently appear in all replicates was considered background). A similar expression pattern was visible at Stages XIX.1 and XX.2 with clear expression in the transition zones (**Figure 4T,U**). *Ov-neuroD* transcripts thus label the transition zones that connect the proliferative lateral lips to the postmitotic central brain.

While our data indicate that the lateral lips are a proliferating region with cells expressing the typical neurogenic transcription factors *Ov-ascl1* and *Ov-ngn*, it is not clear whether these represent different progenitor types. In addition, it was not proven yet that *Ov-neuroD* effectively labeled postmitotic cells. In a hybridization chain reaction experiment combined with immunohistochemistry, we show that *Ov-ngn* and *Ov-ascl1* are likely expressed in a different subset of progenitor cells, since their expression was not overlapping (**Figure 5A–C**). Furthermore, co-staining with the PH3 antibody demonstrated that *Ov-ascl1*$^+$ progenitor cells seem more proliferative compared to *Ov-ngn*$^+$ progenitors, that rarely overlap with the PH3$^+$ population at this late-organogenesis stage (**Figure 5A–G**). The proximity of *Ov-ngn*$^+$ cells to dividing *Ov-ascl1*$^+$ progenitors could be a sign of asymmetric progenitor division during neurogenesis in octopus. In addition, *Ov-neuroD* expressing cells did not co-localize with PH3 immunoreactive cells, indicating that *Ov-neuroD* is absent from mitotically active cells in octopus embryos, at least at Stage XV.2 (**Figure 5H–M**). If the progenitor cells in the lateral lips identified in this study would contribute neurons to the CNS, postmitotic neurons would need to travel long distances from the lateral lips to the central brain. However, direct proof of such neuronal cell migration is still lacking.

## Long distance neuronal migration from spatially patterned lateral lips to the developing brain

In order to map the progeny of cells generated in the lateral lips, we performed lineage tracing experiments using the fluorescent dye CFDA-SE. This technique has been applied in vivo to study temporal neurogenesis patterns in the mammalian cerebral cortex because of the short lifetime of the dye when not incorporated in cells and thus high temporal specificity (**Govindan et al., 2018**). We labeled different populations of cells in the lateral lips in early, mid or late organogenesis phases

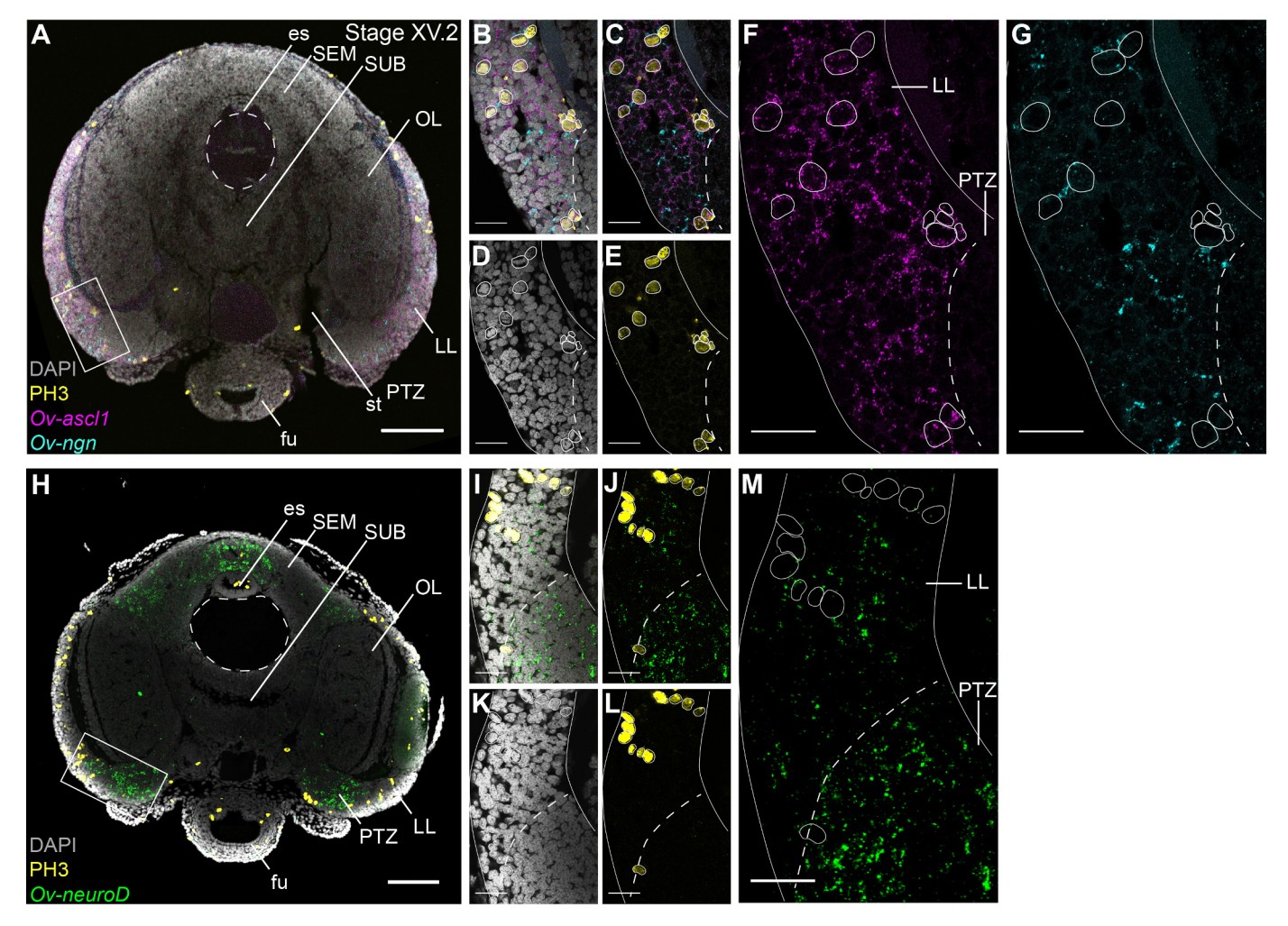

**Figure 5.** Cell proliferation profiles of the progenitor populations in the lateral lips. Multiplex in situ hybridization (HCR v3.0) combined with immunostaining against PH3. (**A**) Overview image showing expression of *Ov-ascl1* and *Ov-ngn* and presence of mitotic cells (PH3+) on a transversal section of a Stage XV.2 embryo. The boxed area covering the lateral lips and posterior transition zone indicates the magnified region in B-G. (**B-G**) Single- and multi-channel magnifications of a single optical section show that *Ov-ascl1* and *Ov-ngn* are expressed in different cell types. In addition, PH3 immunoreactivity is more common in *Ov-ascl1* expressing cells compared to *Ov-ngn* expressing cells. (**H**) Overview image showing expression of *Ov-neuroD* and presence of mitotic cells on a transversal section of a Stage XV.2 embryo. The boxed area covering the lateral lips and posterior transition zone indicates the magnified region in I-M. (**I-M**) Single- and multi-channel magnifications of a single optical section show that *Ov-neuroD* is broadly expressed in the PTZ and does not co-localize with dividing cells in the lateral lips. Scale bars represent 100 µm in A,H and 20 µm in B-G, I-M. Abbreviations as in *Figure 1*.

(Stages IX, XII.1, or XV.2) and traced their progeny to the maturation phase (Stage XIX.2). First, we will focus on the supra- and subesophageal masses (*Figure 6*). At hatching, the major lobes described for the adult supra- and subesophageal masses can be distinguished, with the supraeso-phageal mass consisting of buccal, inferior frontal, superior frontal, subfrontal, vertical, subvertical, basal, dorsal basal and medial basal lobes, and the subesophageal mass consisting of the palliovisceral and pedal lobes (*Figure 6A–C*; *Young, 1971*). For these regions, we identified progenitor cells in the lateral lips that generate output to the different lobes in the supra- and subesophageal masses (*Figure 6D–J*). Specifically, a major contribution to the palliovisceral lobe found its origin in progenitor cells in the dorsal-posterior quadrant of the lateral lips at Stage IX, as was suggested but not empirically proven by *Koenig et al., 2016* (example in *Figure 6D–E*). Other ventral-posterior progenitor populations generated a limited output to the palliovisceral lobe (orange and red injection spots in *Figure 6D,G,I*). In addition, progenitor cells in the posterior lateral lip distinctly contributed

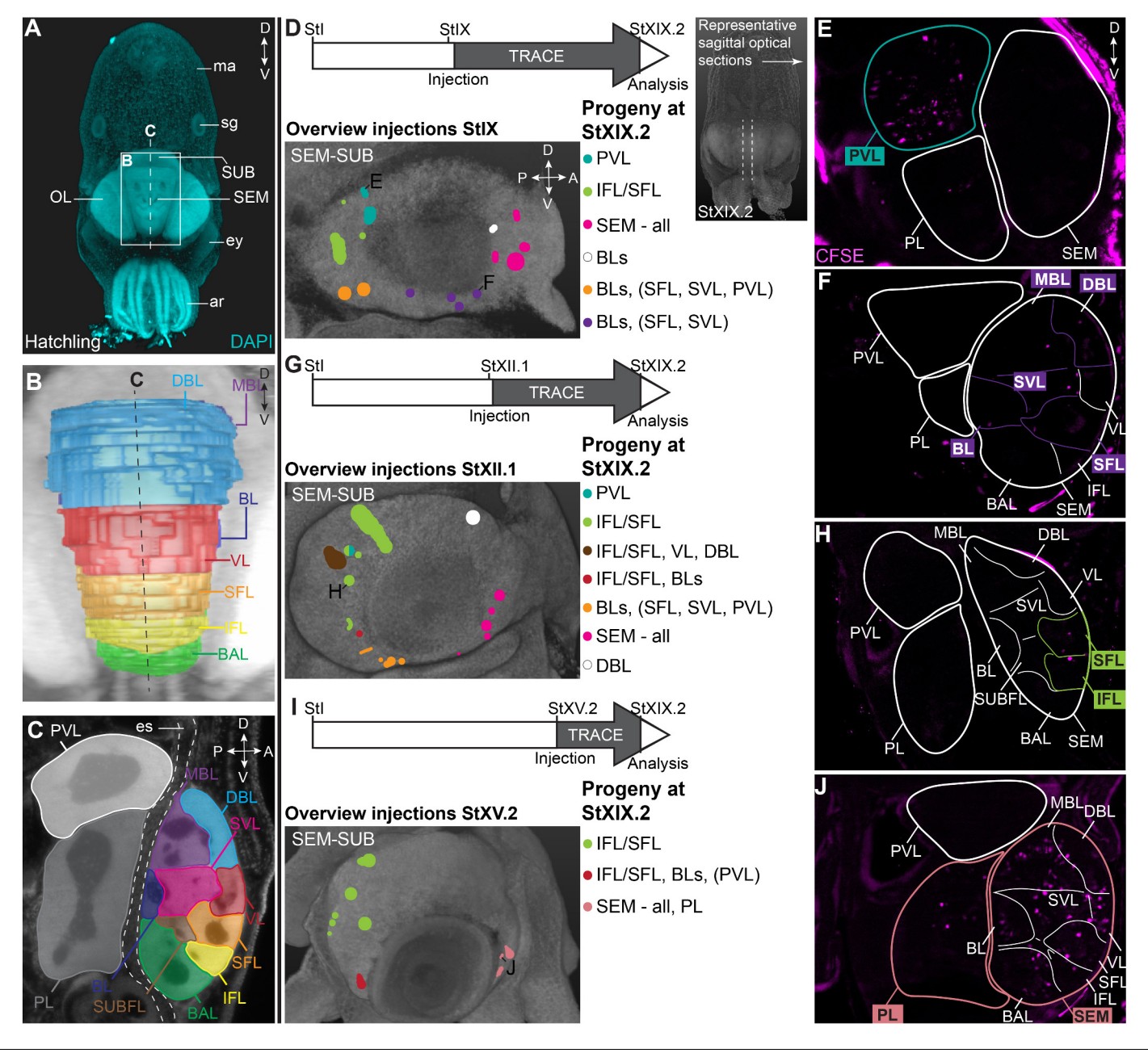

**Figure 6.** CFDA-SE lineage tracing from the lateral lips to the supra- and subesophageal mass. (A) Maximum projection after DAPI staining of a hatchling from the anterior side. Boxed area indicates the magnified region in B and dashed line indicates optical sectioning plane for C. (B-C) Reconstruction of the different lobes in the SEM and SUB of a hatchling in 3D from the anterior side (B) and on an optical section (C). (D-J) Injection of CFDA-SE in the lateral lips at Stage IX (D), Stage XII.1 (G) or Stage XV.2 (I) and tracing until Stage XIX.2 resulted in labeled cells (CFSE positive) in specific brain regions, depending on the location of the progenitor domain. Panels on the left show the location of the CFDA-SE injection site in the later lips, with each colored domain representing a single experimental condition. Progenitor cells in domains with the same color generated comparable output to the brain at Stage XIX.2. On the right, panels E,F,H,J display representative optical sections through the central brain, showing the differential output related to the position of the labeled progenitor population. Targeted regions are indicated in the color corresponding to the color-coded progenitor populations in D,G,I. A, anterior; ar, arm; BAL, buccal lobe; BL, basal lobe; BLs, basal lobes (BL, DBL, MBL); D, dorsal; DBL, dorsal basal lobe; es, esophagus; ey, eye; IFL, inferior frontal lobe; MBL, medial basal lobe; OL, optic lobe; P, posterior; PL, pedal lobe; PVL, pallioviscal lobe; SEM, supraesophageal mass; SFL, superior frontal lobe; sg, stellate ganglion; SUB, subesophageal mass; SUBFL, subfrontal lobe; SVL, subvertical lobe; V, ventral; VL, vertical lobe.

cells to the inferior and superior frontal lobes of the supraesophageal mass at all stages, which has not been reported before by *Koenig et al., 2016* (example in *Figure 6G–H*). Progenitor cells in the ventral lateral lips also produced cells destined to the basal lobes (basal lobe, dorsal basal lobe, medial basal lobe) of the supraesophageal mass (orange and red injection spots in *Figure 6D,G,I*). The majority of cells located in the supraesophageal mass, however, were derived from ventral-anterior progenitor cells in the lateral lips (example in *Figure 6D,F,I,J*). In contrast to the supraesophageal mass and the pallioviseral lobe, few labeled progenitor populations generated cells for the pedal lobe of the subesophageal mass. Apart from the ventral-anterior progenitor populations at Stage XV.2 (example in *Figure 6I–J*), we did not identify progenitor populations that gave rise to a significant number of cells that migrated to the pedal lobe. Occasionally, we identified few, single randomly dispersed cells in the pedal or pallioviseral lobes originating from more ventrally located progenitor populations. Considering the very low number of cells compared to the major output from those progenitors, these cells were not depicted in the overview.

Focusing on the output to the optic lobe and peduncle complex (*Figure 7*), we identified progenitor cells in the dorsal-anterior quadrant of the lateral lips that gave rise to cells in the optic lobes, for which labeled cells generally located to the inner and outer granular layers of the cortex (black populations with an asterisk in *Figure 7A,C,E*; example of progeny in *Figure 7D*). Progenitor cells in the posterior lateral lips at Stages IX and XII.1 generated optic lobe cells that resided in the medulla, while at Stage XV.2, more cells located to the optic lobe cortex as well (example in *Figure 7A,B,E, F*). Ventral-anterior lateral lip progenitors did not generate optic lobe cells. We also identified a clear spatial patterning of progenitors that generate cells for the peduncle complex (olfactory and peduncle lobe). Progenitors in the posterior and ventral lateral lips generated cells destined to this complex (example in *Figure 7A–B*), while populations on the dorsal-anterior side did not. Taken together, our lineage tracing study identified spatial and temporal patterning in the lateral lips, which generate neurons for specific brain regions.

To determine the trajectory that the progeny of lateral lip cells is taking before entering the brain, we performed a short-term lineage tracing study. Hereto, populations of cells in the lateral lips were labeled with CFDA-SE at Stage XIV. Embryos were then allowed to grow for 48–72 hr (reaching Stage XV.2) at which point they were fixed, cleared and imaged with a light sheet microscope to map the location of CFSE positive cells (*Figure 8A*, *Figure 8—figure supplement 1*). We then manually tracked the labeled cells, and reconstructed their trajectory that revealed a continuous stream of cells starting in the lateral lips, passing the posterior (and dorsal) side of the lateral lips and the posterior transition zone, before entering the optic lobe (*Figure 8B,C*). On a series of optical sections, the trajectory can be followed in 2D (*Figure 8D–R*). Labeled cells in the lateral lips (intense labeling, *Figure 8G–L*) divided and migrated posteriorly and entered the posterior transition zone (labeling intensity decreased, *Figure 8D–F*). Then, cells could be traced towards the ventral side of the embryo in the posterior transition zone (*Figure 8F–P*) after which they occupied all layers in the optic lobe (*Figure 8G–R*). Forty-eight hr tracing combined with HCR on thin sections showed migrating cells in the posterior transition zone that expressed *Ov-neuroD* and low-level *Ov-elav*, indicative of newly formed neurons (*Figure 8—figure supplement 2A–D*). The first cells that reached the optic lobe expressed *Ov-elav*, confirming their neuronal identity (*Figure 8—figure supplement 2B,E*). Populations labeled at a different location in the lateral lips showed similar trajectories (even the most anterior labeled population in *Figure 8—figure supplement 1*), with cells passing the dorsal and posterior lateral lips before entering the posterior transition zone and then the optic lobe. Taken together, cells destined for the optic lobe seem to take a defined path via the posterior transition zone.

## Discussion

This study showed that the embryonic octopus brain is practically devoid of dividing cells from Stage XI onwards. Instead, we identified a transient embryonic structure surrounding the developing eye – the lateral lips – that harbors proliferative cells expressing conserved pro-neural transcription factors. We further delineated embryonic neurogenesis in *O. vulgaris* using neural progenitor, pan-neuronal and differentiated-neuron marker genes with conserved expression in neurogenesis, neural specification and differentiation in protostomes and deuterostomes. The spatiotemporal expression patterns of these neurogenic genes suggest their involvement in regulating the development of the CNS in

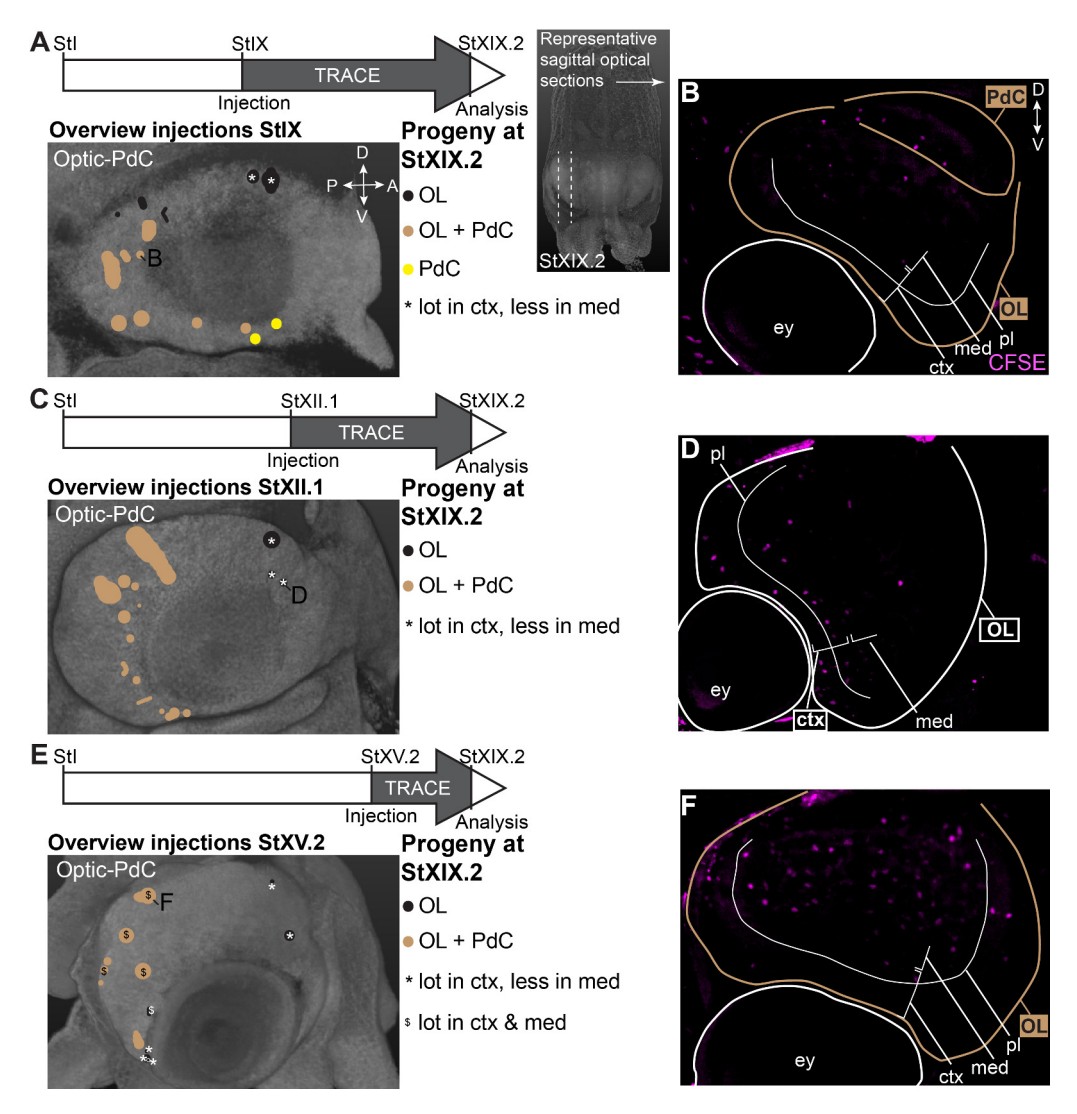

**Figure 7.** CFDA-SE lineage tracing from the lateral lips to the optic lobe and peduncle complex. Injection of CFDA-SE in the lateral lips at Stage IX (**A**), Stage XII.1 (**C**) or Stage XV.2 (**E**) and tracing until Stage XIX.2 resulted in labeled cells in specific regions of the optic lobe and peduncle complex, depending on the location of the progenitor domain. Panels on the left show the location of the CFDA-SE injection site in the later lips, with each colored domain representing a single experimental condition. Progenitor cells in domains with the same color generated comparable output to the brain at Stage XIX.2. On the right, panels B,D,F display representative optical sections through the optic lobe, showing the differential output related to the position of the labeled progenitor population. Targeted regions are indicated in the color corresponding to the color-coded progenitor populations. Abbreviations as in *Figure 6*; ctx, cortex; IGL, inner granular layer; med, medulla; OGL, outer granular layer; PdC, peduncle complex; pl, plexiform layer.

*O. vulgaris*. Koenig et al. have suggested a basic fate map of the neural primordia in *D. pealeii* using DiI tracing experiments, and identified contributions of cells from the lateral lips to the developing central brain (*Koenig et al., 2016*). Our data using a carboxyfluorescein ester that is less leaky compared to DiI, indicate that the spatial map seems to be conserved in cephalopods. In addition, we show that patterning is established at early neurogenesis stages and is grossly temporally maintained, suggesting that lobe-specificity is determined early on within the progenitor area. Next to clear spatial patterning in the lateral lips, we identified long-distance migration of cells generated in the lateral lips towards the central brain in octopuses.

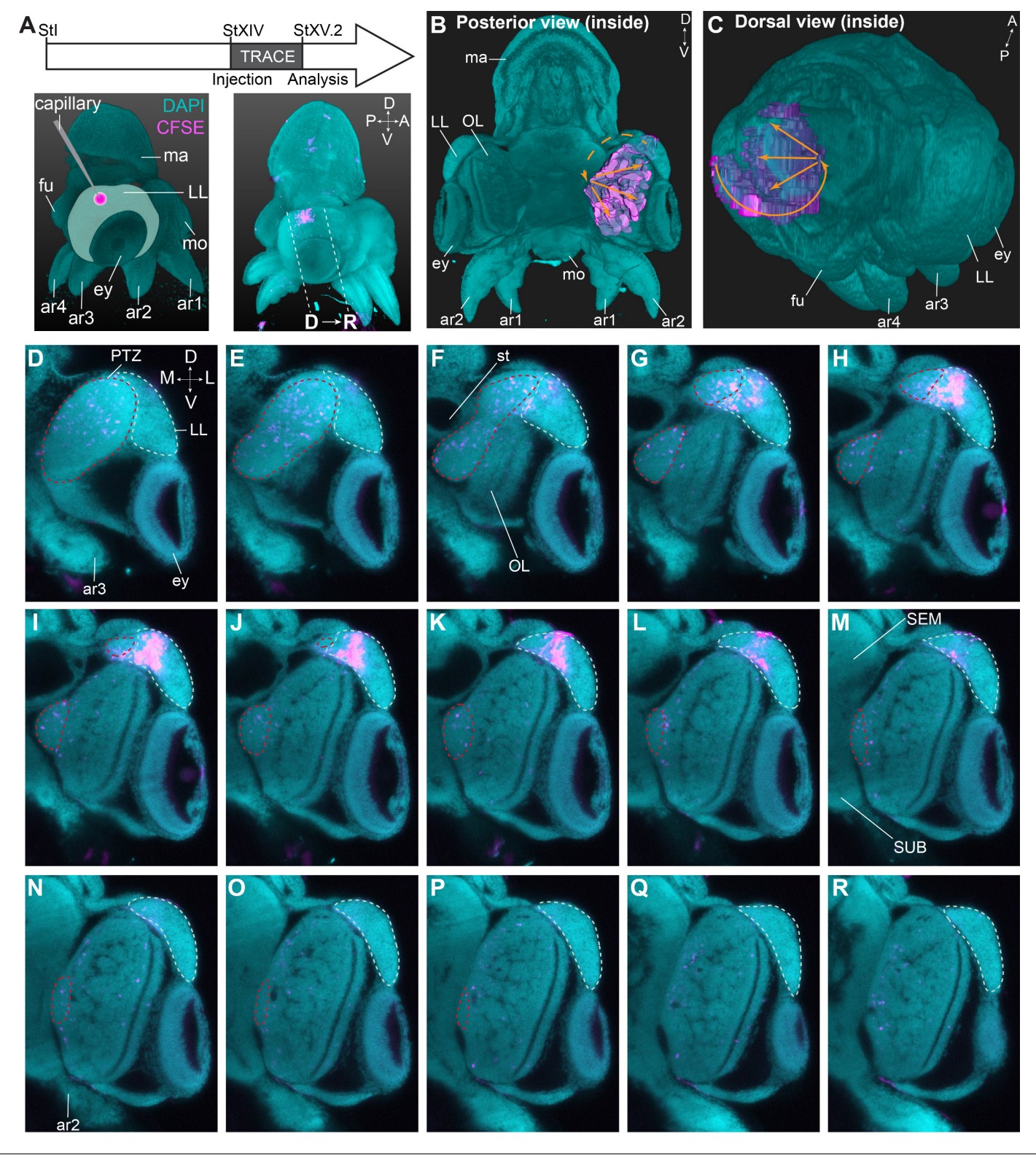

**Figure 8.** Trajectory mapping of migrating cells from the lateral lips to the central brain. (**A**) Experimental setup showing CFDA-SE injection in the lateral lip at Stage XIV and embryo sampling at Stage XV.2. Dashed lines indicate optical sectioning planes for D-R. (**B-C**) Volumetric rendering of the embryo (DAPI, cyan) and the manually traced CFSE object (pink-purple), showing the trajectory (orange arrows) from the injected area in the lateral lips to the optic lobe. (**D-R**) Optical sections through the optic lobe from posterior (D) to anterior (R). The population of labeled cells in the lateral lips is

*Figure 8 continued on next page*

*Figure 8 continued*

visible in G-L. The progeny enters the optic lobes via the posterior transition zone. The lateral lips are encircled in mint green and the posterior transition zone in red. See *Figure 8—figure supplement 1* for an overview of injections resulting in a similar trajectory of cells through the PTZ and *Figure 8—figure supplement 2* showing that migrating cells in the PTZ are neurogenic. Abbreviations as in *Figure 1*; L, lateral; M, medial.

The online version of this article includes the following figure supplement(s) for figure 8:

**Figure supplement 1.** Summary of short-term CFDA-SE lineage tracing.
**Figure supplement 2.** Migrating cells through the PTZ are neurogenic.

## *Octopus vulgaris* CNS maturation is delayed compared to other cephalopods

To determine when and where the first postmitotic immature neurons and differentiated neurons are formed in *O. vulgaris* embryos, we studied the expression of *Ov-elav* and *Ov-syt*, respectively. In the organogenesis phase, *Ov-elav* is expressed in embryonic cells lining the yolk envelope that form the cords, as in *O. bimaculoides* (*Shigeno et al., 2015*). The expression pattern is also consistent with the description of brain precursor regions proposed by Marquis after cytological studies in *O. vulgaris*, reinforcing the use of *elav* as a reliable marker for young octopus neurons (*Marquis, 1989*). We observed low level expression in all cords at Stage IX. In *O. bimaculoides*, expression was already reported from Stage VII.2 onwards, pointing toward rapid neuronal differentiation (*Shigeno et al., 2015*). In contrast to both octopus species, *Sof-elav1* expression in the cuttlefish *Sepia officinalis* is unequally distributed over the different cords, and disappears toward the end of embryonic development, pointing to a different timing of neuronal differentiation and a more advanced maturation of the *Sepia* brain at hatching (*Buresi et al., 2013*). Consistent with this, *S. officinalis* embryos have been described to respond to tactile and chemical as well as visual cues from within the egg capsule from Stage XV.1 and XVI onwards, respectively (*Romagny et al., 2012*) (for interspecies stage comparison, see *Deryckere et al., 2020*). In our hands, *O. vulgaris* embryos only seem to react to visual stimuli (chromatophore contraction in response to light change) and mechanical stimuli (mantle contraction after tapping the chorion) from Stage XIX.1 onwards (preliminary observation). Furthermore, neural processes visualized with immunostaining against acetylated alpha-tubulin revealed the presence of neurites in the cerebral, palliovisceral, pedal, and optic cords of *O. bimaculoides* and *S. officinalis* embryos as early as Stage VIII, while we only observed expression of *Ov-syt* and presence of acetylated alpha-tubulin in the central brain from Stages XIII and XV.2 onwards, respectively (*Baratte and Bonnaud, 2009*; *Shigeno et al., 2015*). These findings further support the delayed maturation of the brain in *O. vulgaris*, that perhaps uses its paralarval phase to complete maturation of the nervous system.

## Model of *O. vulgaris* neurogenesis and the specification of neural cell types

*Ov-soxB1* expression is not restricted to neurectodermal progenitor regions. Consistent with *Sof-soxB1* expression in *S. officinalis*, *Ov-soxB1* is expressed at high level in the surface ectoderm, in the developing eyes, and is absent from the gills and stellate ganglia (*Focareta and Cole, 2016*). Next to (neur)ectodermal expression, *soxB1* is also expressed in sensory epithelia in vertebrates, invertebrate mollusks and acoelomate worms (*Focareta and Cole, 2016*; *Guo et al., 2010*; *Kiernan et al., 2005*; *Le Gouar et al., 2004*; *Neves et al., 2007*; *Semmler et al., 2010*). In addition, SOXB1 proteins are present in both neurectodermal stem cells and differentiated neurons in certain species. *Drosophila soxN* and *Schmidtea polychroa soxB1* for example are expressed in early specification events in the CNS, but also in differentiated parts where they are involved in neuronal differentiation and axonal patterning, suggesting a dual role for protostome soxB1 (*Ferrero et al., 2014*; *Girard et al., 2006*; *Monjo and Romero, 2015*; *Phochanukul and Russell, 2010*). While such a general function in neuronal differentiation of vertebrate soxB1 factors has not been shown, some subtypes of (inter)neurons do require SOXB1 proteins for proper specification and migration (*Cavallaro et al., 2008*; *Ekonomou et al., 2005*; *Panayi et al., 2010*). Similar to many other species, our expression data suggest a dual role for *Ov-soxB1*, in early nervous system development to specify neural fate, and later on to steer neural cell differentiation.

After neurectoderm establishment and neural stem cell formation regulated by SOXB1 transcription factors, neural progenitors must be specified. In most bilaterians, this function has been attributed to members of the bHLH family of transcription factors, including *atonal*, *neurogenin*, *neuroD* and *achaete-scute* subfamilies. Our study is the first one mapping the expression of *ascl1* in cephalopods and together with the expression pattern of *Ov-ngn*, suggests the presence of a neurogenic

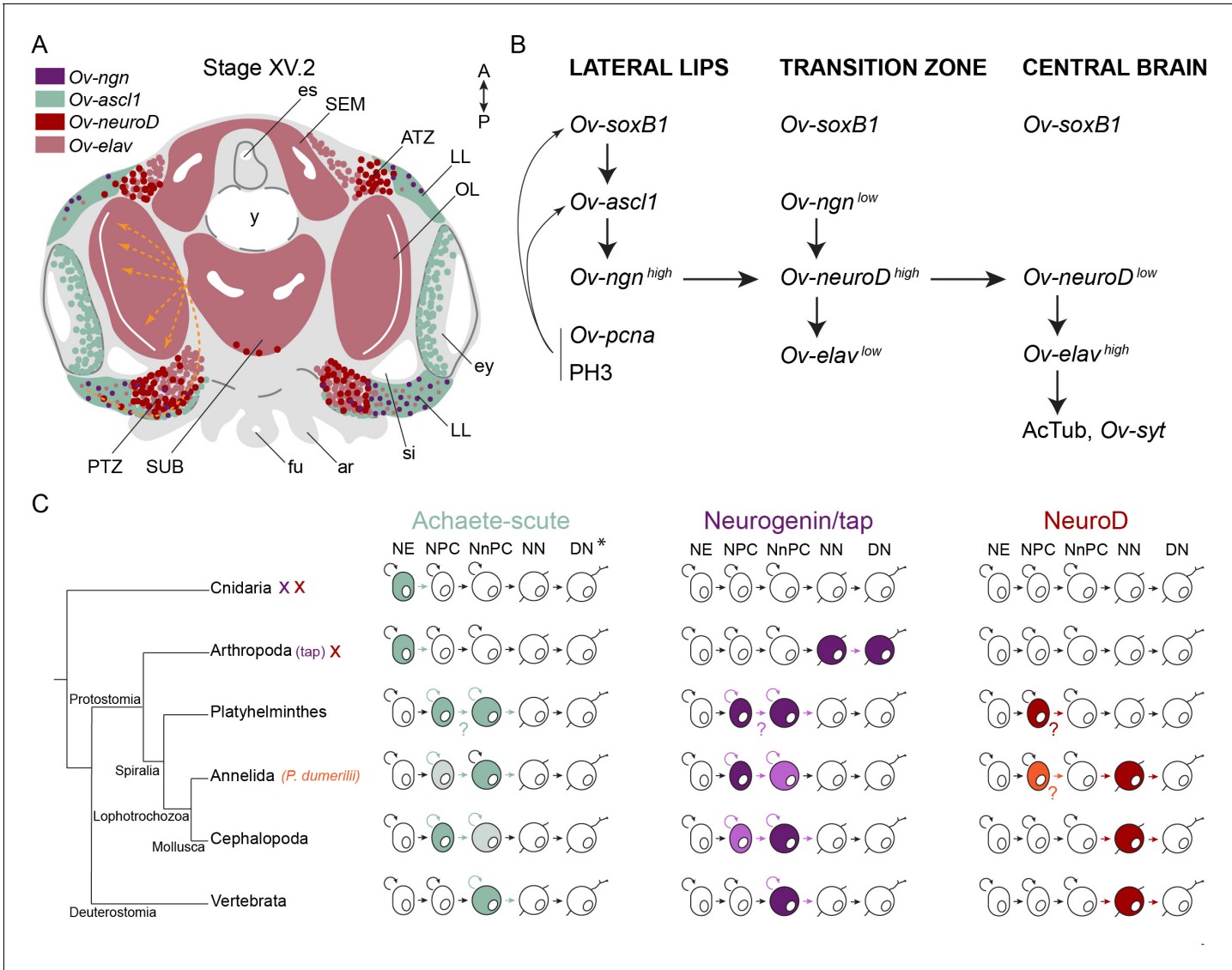

**Figure 9.** Overview of the expression of neurogenic genes and hypothetical neurogenesis process in *O. vulgaris*. (**A**) The expression domains of *Ov-ngn* (purple), *Ov-ascl1* (green), *Ov-neuroD* (red), and *Ov-elav* (pink) are depicted during *O. vulgaris* neurogenesis. Areas indicate high level expression in most cells, whereas dots represent lower expression or high expression in a couple of cells. The orange arrows depict the trajectory taken by cells originating in the lateral lips, passing through the posterior transition zone before entering the optic lobe, as observed after CFDA-SE lineage tracing. The arrows are dashed, considering their 3D projection on a 2D figure. (**B**) *Ov-soxB1*, *Ov-ngn*, and *Ov-ascl1* are all expressed at high level in the lateral lips. In this structure as well, proliferating cells (*Ov-pcna* expressing or PH3 positive) are abundant. They mostly colocalize with *Ov-ascl1* expressing cells that potentially also express *Ov-soxB1*. Our data suggest the onset of differentiation in the transition zones, with low-level expression of *Ov-ngn*, high-level expression of *Ov-neuroD* and, low-level expression of *Ov-elav*. We suggest that the induction of *Ov-neuroD* expression is guided by *Ov-ngn* and that differentiating cells express *Ov-neuroD* before *Ov-elav*. Once arrived in the central brain, neurons start forming synapses (*Ov-syt* expression, presence of acetylated alpha-tubulin). (**C**) Evolutionary comparative expression mapping based on a generalized neural specification and differentiation sequence. See main text for details and references. *Note that the depicted cell types are a generalized state, and that certain phyla/subphyla/classes might lack one or more progenitor types. ar, arm; ATZ, anterior transition zone; DN, differentiated neuron; es, esophagus; ey, eye; fu, funnel; LL, lateral lips; NE, neurectodermal cell; NN, newborn neuron; NnPC, neuronal progenitor cell; NPC, neural progenitor cell; OL, optic lobe; PTZ, posterior transition zone; SEM, supraesophageal mass; si, sinus ophthalmicus; SUB, subesophageal mass; y, yolk.

domain in the lateral lips, outside the cords of the central brain (*Figure 9A,B*). In the annelids *Capitella teleta* and *P. dumerilii*, *neurogenin* and *ash1* are both expressed in actively proliferating neural progenitor cells in the neurectoderm (*Figure 9C*). While *Ct-ngn* positive cells stay on the apical side, the *Ct-ash1*-positive progenitors ingress and undergo limited division before becoming *Ct-elav* positive (*Demilly et al., 2013*; *Meyer and Seaver, 2009*; *Simionato et al., 2008*; *Sur et al., 2017*; *Sur et al., 2020*). In *O. vulgaris*, the *Ov-ascl1* progenitor population seems to be more proliferative compared to the *Ov-ngn* progenitor population, at least at Stage XV.2 (late organogenesis), which suggests their sequential expression could be turned around in cephalopods (*Figure 9B,C*). In addition, the *Ov-elav* expressing neurons in the cords are not located immediately basal from the *Ov-ascl1* or *Ov-ngn* positive pool of neural progenitor cells. In particular, a population of *Ov-neuroD* expressing cells is found in the transition zones, in between the *Ov-ascl1*$^+$ and *Ov-ngn*$^+$ lateral lips and the developing *Ov-elav*$^+$ brain, suggesting an intermediate, *Ov-neuroD*$^+$ population is present in *O. vulgaris* (*Figure 9A,B*). While lost in *Drosophila* and *Ciona intestinalis*, *neuroD* genes can be widely found in bilaterians, where they steer the differentiation of neurons (*Ledent et al., 2002*; *Stollewerk and Simpson, 2005*). In contrast, in the trunk of the annelid *P. dumerilii* (but not *C. teleta*) and in the developing nervous system of the planaria *S. polychroa* and *S. mediterranea*, *neuroD* seems broadly expressed in the neurectoderm, which suggests that the role of neuroD might not be conserved in Spiralia (*Figure 9C*; *Cowles et al., 2013*; *Meyer and Seaver, 2009*; *Monjo and Romero, 2015*; *Simionato et al., 2008*; *Sur et al., 2020*). However, co-localization studies of *neuroD* with progenitor marker genes are still lacking in these species. In vertebrates, *ascl1* and *ngn* are expressed in a complementary fashion, both exerting proneuronal functions in neurogenesis (*Bertrand et al., 2002*; *Castro et al., 2011*; *Farah et al., 2000*; *Gradwohl et al., 1996*; *Lee et al., 1995*; *Ma et al., 1996*). Similar to our data in octopus, *neurogenin* expression precedes, but also partially overlaps with that of *neuroD* (*Grimaldi et al., 2008*). In contrast, *Drosophila* has only one *neurogenin/neuroD* homolog called *tap*, which relates better to *neurogenin* based on sequence similarities, but does not have a proneural role and is expressed in a few neurons, regulating their differentiation, axonal growth and guidance (*Gautier et al., 1997*; *Vervoort and Ledent, 2001*; *Yuan et al., 2016*). In arthropods, the members of the *achaete-scute* subfamily are expressed in proneural clusters and promote the generation of neural progenitor cells from quiescent ectodermal cells (*Bertrand et al., 2002*; *Cubas et al., 1991*; *Quan and Hassan, 2005*; *Skeath and Carroll, 1991*; *Stollewerk and Chipman, 2006*). In cnidaria as well, *ashA* promotes neurogenesis during its development (*Layden et al., 2012*; *Figure 9C*). In *O. bimaculoides*, *neurogenin* and *neuroD* transcripts were detected in the prospective cerebral, palliovisceral and pedal cords at Stage VIII, based on whole mount in situ hybridization, but did not distinguish the lateral lips from the cordal areas (*Shigeno et al., 2015*). Based on cross-sections, *neurogenin* transcripts were found on the surface of the embryo, and *neuroD* more at the level of the cords, similar to our observations in *O. vulgaris* (*Shigeno et al., 2015*). In the vertebrate neural tube, NEUROD was also found on the basal side, in postmitotic neuronal and glial cells and is required for the differentiation of neurons in the inner ear, cerebellum and hippocampus (reviewed in *Dennis et al., 2019*). Its expression pattern in *O. vulgaris* suggests that upon differentiation, neurons express *Ov-neuroD* before expressing *Ov-elav*, which is similar to the role of *Ov-neuroD* as neuronal-differentiation bHLH transcription factor, as described for vertebrates and the annelid *C. teleta* (*Bertrand et al., 2002*; *Farah et al., 2000*; *Lee et al., 1995*; *Sur et al., 2020*). We also revealed possible overlapping expression of *Ov-soxB1* with *Ov-ascl1* and *Ov-ngn* in the lateral lips throughout development, suggesting a similar activation of *Ov-ascl1* and *Ov-ngn* by *Ov-soxB1* in the presumptive neurectoderm (*Figure 9B*; *Amador-Arjona et al., 2015*; *Ferrero et al., 2014*). Therefore, *Ov-ngn* and *Ov-ascl1* are expressed at the right time and place to be the major proneuronal genes for the formation of the central nervous system in *O. vulgaris*. Our data further substantiate the conserved expression of proneural bHLH transcription factors, which suggests they might already have been present in the ur-bilateria.

## Spatial patterning of the lateral lips and long-distance migration to the central brain

The lateral lips harbor proliferating cells with a signature of neural progenitors, while mature neurons are found in the cords. The overt absence of proliferative cells in the developing brain in an organism with such a big and centralized CNS is striking, but not uncommon for mollusks. Mitosis in the *Aplysia* CNS is infrequent from early embryogenesis to adulthood (*Jacob, 1984*). Compared to other

invertebrates with large brains such as insects, however, this seems to be a rather unique strategy. The developing optic lobe in *Drosophila* for example has dividing neuroblasts in two main proliferation centers inside the lobe. These neuroblasts had previously invaginated from the neuroepithelium and start dividing only when in the brain (*Álvarez and Díaz-Benjumea, 2018*; *Apitz and Salecker, 2015*; *Green et al., 1993*; *Hofbauer and Campos-Ortega, 1990*; *Walsh and Doe, 2017*). The physical disconnection between the neural stem cells and their progeny in octopus suggests that secondary progenitor cells or neurons migrate towards the central brain where they integrate.

In lophotrochozoans such as the annelids *C. teleta* and *P. dumerilii,* dividing neural progenitors are located on the apical side of the neuroepithelium and it is their progeny that ingresses to form the CNS (*Meyer and Seaver, 2009*; *Monjo and Romero, 2015*; *Sur et al., 2020*). While resembling the postmitotic migration in *O. vulgaris*, their migratory path is short and the number of neurons remains limited, making direct comparison with *O. vulgaris* difficult. As described here and by Shigeno et al., the nervous system of cephalopods originates from a system of cords, similar to the nervous system of primitive mollusks like Aculifera including chitons, and in contrast to Concifera including gastropods, whose CNS knows a ganglionic origin (*Richter et al., 2010*; *Shigeno et al., 2015*; *Sumner-Rooney and Sigwart, 2018*). Gastropod ganglia seem to derive from ectodermal cells in the body wall that proliferate and delaminate/migrate inwards to join the developing ganglia. These studies suggest that similar to annelids, neural progenitor cells in the gastropod mollusk *Aplysia californica*, divide in proliferative zones in the body wall and their progeny migrates few cell lengths to the nearest ganglion (*Demian and Yousif, 1975*; *Jacob, 1984*). Depending on the location in the proliferative zone, cells migrate either in a columnar stream, or individually using pseudopodia. Evidence for cell migration in cephalopods came from *Loligo vulgaris*, where in vitro cultures of the 'oculo-ganglionar complex' showed extensive migration of cells and differentiation into bi-and multipolar neurons (*Marthy and Aroles, 1987*). Here, we showed that the progeny of octopus lateral lip progenitors migrates over long distances before integration in the CNS. Strikingly, we observed that the migratory path does not just represent the shortest route to destiny. Instead, we found that independent from the anterior-posterior axis, progenitor populations in the dorsal lateral lips generate cells for the optic lobe that pass through the posterior transition zone. Cells thus migrate from the most anterior part towards the posterior part of the lateral lips before entering the posterior transition zone and eventually migrating in all directions, spreading throughout the optic lobe. This suggests active, directed migration controlled by guidance cues. Which cell intrinsic and/or extrinsic cues govern the migratory process, remains to be studied.

Active neural migration guided by extrinsic cues is common in vertebrates that build their brain from a large, spatially patterned and proliferating neural epithelium folded into a tube. Neurons are generated in a temporally controlled fashion and migrate away from the progenitor zone surrounding the ventricles to form the grey matter. The latter entirely consists of postmitotic neurons that start growing neurites and connect while additional postmitotic neurons are migrating in between them to their target region (*Marín et al., 2010*; *Marín and Rubenstein, 2003*; *Paridaen and Huttner, 2014*; *Taverna et al., 2014*). In that respect, the development of the *O. vulgaris* brain seems strikingly similar. Yet *O. vulgaris* seems to pattern the lateral lips in such a manner that entire brain lobes are derived from specific areas in the lateral lips, whereas specification in vertebrates links spatial (i.e. progenitor area) or temporal patterning to cell types that are intermingled in certain regions. An example is the subpallium, that generates (among other cell types) cortical interneurons, while the dorsal pallium generates pyramidal neurons, and both are mixed in the cerebral cortex. Another example is the postnatal V-SVZ, where different areas generate distinct interneuron types destined for the olfactory lobe (*Hatten, 1999*; *Marín and Rubenstein, 2001*; *Marín and Rubenstein, 2003*; *Medina and Abellán, 2009*).

Neurons that are born in a specific region of the lateral lips thus seem instructed to migrate to a certain brain lobe, but how they get specified to different cell types within that lobe remains unclear. Indeed, cells from one injection site mostly spread out over a whole lobe, except for the optic lobe, where we identified lateral lip subregions that are biased to generate optic lobe cortex or optic lobe medulla neurons. Interestingly, the proportion of cells in those layers increased when progenitor labeling was performed at later stages, indicating that cell types might be specified in a temporal manner as well. To investigate this observation in more detail, additional molecular markers for different neuronal subpopulations need to be identified. Furthermore, while proneural bHLH transcription factors are likely involved in specifying and maintaining neural progenitor identity, the

transcription factors that serve as terminal selectors need to be revealed. Lastly, as the lateral lips shrink to almost completely disappear towards the end of development, there must be a second, yet undefined source of progenitors, which generates additional neurons in post-hatching stages for the adult brain.

# Materials and methods

**Key resources table**

| Reagent type (species) or resource | Designation | Source or reference | Identifiers | Additional information |
|---|---|---|---|---|
| Antibody | Anti-Tubulin, Acetylated antibody (Mouse Monoclonal) | Sigma | Cat#: T6793 | (1:300) |
| Antibody | Anti-phospho-Histone H3 (Ser10) Antibody (Rabbit Polyclonal) | Millipore | Cat#: 06–570 | (1:300) |
| Antibody | Donkey anti-Mouse IgG (H+L) Secondary Antibody, Alexa Fluor 488 | Life Tech (Invitrogen) | Cat#: A-21202 | (1:300) |
| Antibody | Donkey anti-Rabbit IgG (H+L) Highly Cross-Adsorbed Secondary Antibody, Alexa Fluor 555 | Life Tech (Invitrogen) | Cat#: A-31572 | (1:300) |
| Antibody | Fluorescein Antibody (Goat Polyclonal) | Novus Biologicals | Cat#: NB600-493 | (1:300 on sections, 1:500 whole mount) |
| Antibody | Donkey anti-Goat IgG (H+L) Cross-Adsorbed Secondary Antibody, Alexa Fluor 488 | Invitrogen | Cat#: A-11055 | (1:300) |
| Strain, strain background (*Escherichia coli*) | JM109 chemocompetent cells | Promega | Cat#: L2005 L1001 | |
| Chemical compound, drug | Mowiol 4–88 | Sigma | Cat#: 81381 | |
| Chemical compound, drug | TRI Reagent Solution | Invitrogen | Cat#: AM9738 | |
| Chemical compound, drug | Eukitt quick-hardening mounting medium | Sigma | Cat#: 03989 | |
| Chemical compound, drug | Poly(dimethylsiloxane-co-methylphenylsiloxane) viscosity 125 cSt | Sigma-Aldrich | Cat#: 378488 | |
| Chemical compound, drug | Mineral oil | Sigma-Aldrich | Cat#: M8410 | |
| Chemical compound, drug | Silicone Elastomer, 2 Part, 1:1 Mix, Sylgard 170, Black / White, Container, 2 kg | DOWSIL | Cat#: 101693 | |
| Commercial assay or kit | RNeasy Micro Kit | Qiagen | Cat#: 74004 | |
| Commercial assay or kit | SMARTer PCR cDNA Synthesis Kit | Takara Bio Inc | Cat#: 634925 | |
| Commercial assay or kit | NEBNext Single Cell/Low Input cDNA Synthesis and Amplification Module | New England BioLabs. | Cat#: E6421S | |
| Commercial assay or kit | Superscript III Reverse Transcriptase | Invitrogen | Cat#: 18080–044 | |

*Continued on next page*

*Continued*

| Reagent type (species) or resource | Designation | Source or reference | Identifiers | Additional information |
|---|---|---|---|---|
| Commercial assay or kit | TOPO TA Cloning Kit, Dual Promoter, without competent cells (25 reactions). | Invitrogen | Cat#: 450640 | |
| Commercial assay or kit | Micro Bio-Spin P-30 Gel Columns, Tris Buffer (RNase-free) | BioRad Lab. | Cat#: 7326250 | |
| Commercial assay or kit | Ribomap Kit | Roche | Cat#: 5266190001 | |
| Commercial assay or kit | Bluemap detection kit | Roche | Cat#: 5266327001 | |
| Other | Proteinase K, recombinant, PCR Grade | Roche | Cat#: 3115887001 | ISH (1:1000) HCR (1:3000) |
| Other | Tissue-Tek Biopsy 6-Chamber Cassette | Sakura | Cat#: 4073 Biopsy 6 Chamber Cassette White 1.000pcs | For paraffin embedding |
| Other | DAPI | Sigma Aldrich | Cat#: 32670–5 MG-F | (1:1000) |
| Other | HCR Amplifier B1, Alexa Fluor 546 | Molecular Instruments (US) | | HCR Amplifier |
| Other | HCR Amplifier B2, Alexa Fluor 647 | Molecular Instruments (US) | | HCR Amplifier |
| Other | HCR Amplifier B3, Alexa Fluor 488 | Molecular Instruments (US) | | HCR Amplifier |
| Other | Glass Capillaries, 3.5', For All Model Nanoject Models | Drummond | Cat#: 3-000-203-G/X | For microinjection |
| Other | CFDA-SE | SanBio | Cat#: 14456–10 | (1 mM for trajectory mapping, 0.1 mM for long term tracing in filtered seawater) |
| Other | Fast Green FGF | Sigma-Aldrich | Cat#: F7252 | (0.3 mg/ml) |
| Software, algorithm | Fiji: ImageJ | DOI: 10.1038/nmeth.2019 | | |
| Software, algorithm | SMRT Link v. 9.0.0 | Pacific Biosciences (PACBIO) | | |
| Software, algorithm | IsoSeq 3.3 | Pacific Biosciences (PACBIO) | | |
| Software, algorithm | ARIVIS Vision4D Zeiss Edition 3.1.4 | ARIVIS AG | | |
| Software, algorithm | Adobe Photoshop | Adobe | | |
| Software, algorithm | Blast2GO | DOI: 10.1093/bioinformatics/bti610 | | |
| Software, algorithm | insitu_probe_generator | DOI: 10.5281/zenodo.4086058 | | |
| Software, algorithm | tBLASTn, BLASTp | NCBI: https://blast.ncbi.nlm.nih.gov/Blast.cgi | | |
| Software, algorithm | UCSC Genome Browser on Euprymna scolopes Euprymna scolopes Assembly (eupSco1) | DOI: 10.1073/pnas.1817322116 | | |
| Software, algorithm | ORFfinder | NCBI (RRID:SCR_016643) | | |
| Software, algorithm | MUSCLE Alignment | DOI: 10.1093/bioinformatics/bth090 | | |
| Software, algorithm | MegaX | DOI: 10.1093/molbev/msy096 | | |
| Software, algorithm | TrimAl | DOI: 10.1093/bioinformatics/btp348 | | |
| Software, algorithm | IQTree | DOI: 10.1093/molbev/msu300 | | |
| Software, algorithm | FigTree v.1.4.4 | DOI:http://tree.bio.ed.ac.uk Software/Figtree/ | | |
| Software, algorithm | CDD Search Tool | DOI: 10.1093/nar/gkz991 | | |

## Animals

Live *O. vulgaris* embryos were obtained from the lab of E. Almansa (IEO, Tenerife), transferred to the lab of Developmental neurobiology and kept in a closed standalone system (*Deryckere et al., 2020*). Embryos were observed, staged and sampled daily, followed by overnight fixation in 4% paraformaldehyde (PFA) in phosphate buffered saline (PBS). After a wash in PBS, embryos were manually dechorionated with tweezers and transferred to embedding cassettes (Tissue-Tek Biopsy 6-Chamber Cassette, Sakura). For paraffin processing, the cassettes were immersed in 0.9% NaCl overnight before progressive dehydration and paraffin-embedding using an Excelsior AS Tissue Processor and HistoStar Embedding Workstation (Thermo Scientific). 6 µm-thick transversal sections were made for subsequent immunohistochemistry or in situ hybridization.

## Immunohistochemistry on paraffin sections

Embryo sections were processed using an automated platform (Ventana Discovery, Roche) for direct fluorescent staining. Primary antibodies mouse anti-Acetylated alpha Tubulin (Sigma T6793), rabbit anti phospho-histone H3 (Ser10) (Millipore 06–570), and goat anti fluorescein (Novus Biologicals NB600-493) and secondary antibodies donkey anti-mouse Alexa 488, donkey anti-rabbit Alexa 555, and donkey anti-goat Alexa 488 (Life Technologies) were each diluted in Pierce Immunostain or antibody diluent (Roche) and incubated at a final concentration of 1:300. Sections were then incubated in DAPI and mounted in Mowiol. Images were acquired with a Leica DM6 upright microscope and minimum/maximum displayed pixel values were adjusted in Fiji (*Schindelin et al., 2012*). Images used in the figures represent the staining pattern observed in multiple embryos (number of replicates presented in *Supplementary file 1*).

## RNA extraction, sequencing, and Iso-Seq data analysis

In order to construct a full-length transcriptome of *O. vulgaris* embryos and paralarval brains, the Iso-Seq method was used. RNA was extracted from a pool of 25 Stage XI-XII embryos using Tri-reagent (Invitrogen) and the Qiagen Micro kit (Qiagen). cDNA was synthesized with the Clontech SMARTer PCR cDNA Synthesis Kit (Takara Bio Inc). RNA was also extracted from dissected brains of one-day old paralarvae in a similar manner and cDNA was synthesized using the NEBNext cDNA Synthesis and amplification kit. Both samples were sequenced on the PacBio Sequel at the Genomics Core at KU Leuven (Belgium) following the protocol recommended by PacBio. Only cDNAs containing polyA-tails were selected, with the aim to retrieve full-length transcripts. This resulted in a total of 12,017,703 subreads for the embryo and 15,426,835 subreads for the brain sample. The raw data files were processed with SMRT Link release 9.0.0 software. The IsoSeq 3.3 pipeline was followed to generate consensus reads (inc polish, min.passes = 1). Lima (-isoseq) was used to retain full-length fragments that possess both primers only, to remove unwanted primer combinations and to orient the sequences. Subsequently, Poly(A) tails were trimmed and concatemers were removed. This resulted in 22,757 and 28,490 high-quality polished isoforms for the embryo and hatchling brain samples, respectively. Data have been deposited in SRA under the following accession number PRJNA718058.

## Identification and cloning of *O. vulgaris* genes

Putative homologs of *O. vulgaris achaete-scute*, *neurogenin*, *neuroD*, *elav*, *soxB1*, *synaptotagmin* and *pcna* genes were identified using tBLASTn searches against the ISOseq transcriptomes. *O. vulgaris* hits hereafter named *Ov-ascl1*, *Ov-ngn*, *Ov-neuroD*, *Ov-elav*, *Ov-soxB1*, *Ov-syt*, and *Ov-pcna* were then blasted against the NCBI database to verify sequence homology. Primers were designed (primer sequences in *Supplementary file 2*) to isolate a 500–1000 bp fragment from mixed-stage *O. vulgaris* embryo cDNA (synthesized using Superscript III Reverse Transcriptase (Invitrogen)) (probe sequences in *Supplementary file 3*). The resulting PCR products were TA cloned into the pCRII-TOPO vector (Invitrogen) and sequenced by LGC Genomics (Berlin). After plasmid linearization, antisense digoxigenin-(DIG) labeled RNA probes were generated using an Sp6- or T7-RNA polymerase and DIG RNA labeling mix (both Roche) following the manufacturer's protocol. The probes were cleaned using Micro Bio-Spin P-30 Gel Columns with RNase-free Tris Buffer (BioRad).

## Colorimetric in situ hybridization

Paraffin sections were processed using an automated platform (Ventana Discovery, Roche) with RiboMap fixation and BlueMap detection kits (Roche) for in situ hybridization. In short, sections are deparaffinated, heated to 37°C, post-fixed and pretreated. Then, a 4 min digestion with proteinase K (Roche, 1:1000 in PBS-DEPC) is followed by probe titration (100–300 ng per slide dependent on the probe, dissolved in Ribohybe reagent (Roche)), denaturation at 90°C for 6 min and hybridization at 70°C for 6 hr. Three stringency washes in 0.1X SSC at 68°C for 12 min each are followed by post-fixation. The anti-DIG-Alkaline phosphatase antibody (Roche) is added and sections are incubated for 30 min after which a colorimetric signal (BCIP/NBT) is developed for 4–9 hr (probe dependent). The tissue is counterstained with Red Counterstain II (Roche), followed by dehydration and mounting using Eukitt quick-hardening mounting medium (Sigma). Bright-field images were taken with a Leica DM6 upright microscope and background was subtracted in Photoshop. Images used in the figures represent the expression pattern observed in multiple embryos (number of replicates presented in *Supplementary file 1*).

## Hybridization Chain Reaction v3.0

HCR-3.0-style probe pairs for fluorescent in situ mRNA visualization were generated for *Ov-ascl1*, *Ov-elav*, *Ov-neuroD* and *Ov-ngn*. Hereto, we used the insitu_probe_generator (*Null and Özpolat, 2020*), followed by BLAST searches using Blast2GO (*Conesa et al., 2005*) to minimize potential off-target hybridization. DNA oPools were ordered from Integrated DNA Technologies, Inc (probe sets in *Supplementary file 4*) and dissolved in DNase/RNase-Free distilled water (Invitrogen). HCR amplifiers with fluorophores B1-Alexa Fluor-546, B2-Alexa Fluor-647, and B3-Alexa Fluor-488 were ordered from Molecular Instruments, Inc. The Molecular Instruments HCR v3.0 protocol for FFPE human tissue sections, based on *Choi et al., 2016* and *Choi et al., 2018* was followed (*Choi et al., 2016*; *Choi et al., 2018*). Described here are adaptations from this protocol. Paraffin sections were baked at 65°C for 30 min and subsequently deparaffinized with Xylene (2 x 4 min) and 100% EtOH (3 x 4 min). To permeabilize the tissue, slides were treated with proteinase K (Roche, 1:3000 in PBS-DEPC) for 5 min at 37°C. Slides were then rinsed 2 x 2 min with autoclaved MQ and immediately processed for HCR. After a 30 min pre-hybridization step, probe solution (0.4 pmol per probe in probe hybridization buffer) was incubated overnight. The next day, 4.5 pmol of hairpin h1 and 4.5 pmol of hairpin h2 were snap-cooled (95°C for 90 s, 5 min on ice followed by 30 min at room temperature) and added to 75 μL of amplification buffer. After overnight amplification, excess hairpins were removed by washing 3 x 10 min with 5X SSCT. After HCR, we proceeded with immunohistochemistry as described above and lastly, sections were incubated in DAPI and mounted in Mowiol. Images were acquired using a confocal microscope (Fluoview FV1000, Olympus) and minimum/maximum displayed pixel values were adjusted in Fiji (*Schindelin et al., 2012*). Autofluorescence in the different channels and potential aspecific amplifier binding were assessed first. Then, all probes were tested individually before multiplexing. HCR was performed on at least three slides per probe, on at least two different embryos.

## Phylogenetic analysis

To determine homology, phylogenetic analyses of ASH, NEUROD, NEUROG, ELAV, and SOX families were performed. Full-length protein sequences (when available) were obtained using BLASTp, tBLASTn or word search in NCBI or from published articles (accession numbers in *Supplementary file 5*). In the case of *E. scolopes*, the BLAST genome server together with peptide sequences on cephalopodresearch.org were used (*Belcaid et al., 2019*). In the case of *O. vulgaris*, nucleotide sequences were obtained using tBLASTn searches against the Iso-Seq transcriptomes, followed by a search for open reading frames using the ORFfinder tool in NCBI, which also provided the translated protein sequences (*O. vulgaris* protein sequences in *Supplementary file 6*). All protein sequences were aligned using the 'MUSCLE Alignment' feature (*Edgar and Sjölander, 2004*) within MEGA-X (*Kumar et al., 2018*). The matrix was then trimmed using TrimAl (*Capella-Gutiérrez et al., 2009*) via the *Automated one* algorithm. The best-fit substitution model for each alignment was determined using the Bayesian information criterion in IQ-TREE (*Nguyen et al., 2015*). Maximum likelihood analyses using the LG+G4 (ELAV), JTT+I+G4 (NEUROD/NGN), VT+I+G4 (ASH), or JTT+G4+F (SOX) substitution models for protein evolution were also performed in IQ-

TREE, with branch supports calculated by 10.000 Ultrafast bootstrap replicates (UFBoot2) from maximum 1000 iterations (stopping rule) (*Hoang et al., 2018*; *Nguyen et al., 2015*). The produced consensus trees were rooted and visualized with FigTree v1.4.4 (*Rambaut, 2018*). Domains in the predicted protein sequences of candidate *O. vulgaris* homologs were identified with NCBIs' conserved domain database (CDD) search tool (*Lu et al., 2020*).

## CFDA-SE injection procedure

For injection of live *O. vulgaris* embryos, a glass capillary (3-000-203-G/X, Drummond), that was pulled using a Laser-Based Micropipette Puller (P-2000, Sutter Instrument; Heat 450, Fil 4, Vel 150), and opened at 30 µm was mounted on a micromanipulator (M3301L, WPI) and connected to a FemtoJet (Eppendorf) via an injection tube. Excess seawater was removed from the egg using a tissue after which the egg was transferred to a Sylgard-coated Petri dish (Sylgard 170, Dowsil). The egg was stabilized with tweezers and the capillary was inserted in the embryonic tissue, from the stalk side through the chorion, in an angle of about 10–30 ° relative to the dorso-ventral axis of the embryo. A single 50–100 nL dose of a carboxyfluorescein diacetate succinimidyl ester (CFDA-SE) working solution (1 mM for trajectory mapping, 0.1 mM for long term tracing in filtered sea water, 3% FastGreen) was injected in the lateral lips anterior, posterior, dorsal or ventral from the eye placode at developmental Stage XIV for trajectory mapping (n = 8) and Stages IX (n = 20), XII.1 (n = 21) and XV.2 (n = 16) for long-term tracing. CFDA-SE is a non-fluorescent and highly membrane-permanent molecule that is cleaved by intracellular esterases once taken up by cells, resulting in a trapped, fluorescent carboxyfluorescein succinimidyl ester (CFSE) molecule that is bound to amino groups of intracellular proteins (*Progatzky et al., 2013*; *Quah et al., 2007*). The dye thus acutely labels all cells present at the injection location, and their progeny due to its long-lasting stability. After injection, embryos were placed back in a Petri dish with filtered sea water. Successful uptake of the dye was verified after 30 min using a fluorescence binocular (SteREO Discovery.V8 with AxioCam MRc5 (Zeiss)). Individual eggs were then incubated in a 96-well plate in filtered sea water, 2% Penicillin/Streptomycin, in a 19°C incubator in the dark (Heratherm IMC18, Thermo Scientific). Viable embryos were sampled after 48 or 72 hr for trajectory mapping or when reaching Stage XIX.2 for long-term tracing, fixed overnight in 4% PFA in PBS and then stored in PBS until dechorionation and clearing or paraffin embedding.

## Clearing and whole mount immunohistochemistry

Dechorionated embryos were cleared before light sheet imaging as previously described (*Deryckere et al., 2020*). Whole mount immunohistochemistry was performed for long term CFDA tracing, during the PBS washing steps in between incubation in Sc*ale*CUBIC-1 and Sc*ale*CUBIC-2 as follows: after clearing in Sc*ale*CUBIC-1, embryos were washed four times; one time for 5 min and three times for 2 hr in PBS supplemented with 0.3% Triton X-100 (PBS-T) and then incubated in primary antibody solution in PBS-T (goat anti-fluorescein (Novus Biologicals NB600-493, 1:500, preincubated overnight with non-injected embryos)) for 2 days at 4°C. Embryos were then washed three times for 2 hr in PBS-T, secondary antibody solution was added (donkey anti-goat Alexa 488 (Life Technologies, 1:300)) and incubated overnight at 4°C after which the samples were washed three times for 2 hr in PBS-T and then incubated in 1/2-water diluted Sc*ale*CUBIC-2 at 37°C.

## Light sheet fluorescence microscopy and analysis

Cleared and stained embryos were glued with their yolk sack on a metal plunger and imaged using a Zeiss Z1 light sheet microscope (Carl Zeiss AG, Germany) in low-viscosity immersion oil mix (Mineral oil, Sigma M8410 and Silicon oil, Sigma 378488, 1:1). Then, 3D reconstructions were generated in Arivis (Vision4D, Zeiss Edition 3.1.4).

For lineage tracing after CFDA-SE injection, the distribution of the progeny was mapped using optical sections. In addition, for trajectory mapping, the region including most CFSE labeled cells was manually traced using the objects drawing tool in Arivis, in order to visualize the trajectory in 3D. This same tool was also used to reconstruct the eye, optic lobe, lateral lips and posterior transition zone at Stage XV.2 for *Figure 1*, and to reconstruct the different lobes in the central brain at hatching for *Figure 6*.

## Acknowledgements

The authors wish to thank Eduardo Almansa (Instituto Español de Oceanografía, Santa Cruz de Tenerife, Spain) and Camino Gestal (Institute of Marine Research, Vigo, Spain) for generously providing *Octopus vulgaris* embryos. We thank the laboratory of Cris Niell (Institute for Neuroscience, University of Oregon, USA) for sharing their Hybridization Chain Reaction protocol and Caroline Zandecki for generating the *Ov-syt* probe. We are also grateful to the Genomics Core Leuven (http://www.genomicscore.be) for the IsoSeq sequencing and Joke Allemeersch for data analysis. We wish to thank Maria Antonietta Tosches for critically reading the manuscript.

## Additional information

### Funding

| Funder | Grant reference number | Author |
|---|---|---|
| Fonds Wetenschappelijk Onderzoek | SB/1S19517N | Astrid Deryckere |
| Fonds Wetenschappelijk Onderzoek | SB/11D4120N | Ali Murat Elagoz |
| KU Leuven | ID-N/20/007 | Eve Seuntjens |
| Stazione Zoologica Anton Dohrn | | Ruth Styfhals |
| KU Leuven | IOFm/17/014 | Gregory E Maes |

The funders had no role in study design, data collection and interpretation, or the decision to submit the work for publication.

### Author contributions

Astrid Deryckere, Conceptualization, Formal analysis, Funding acquisition, Investigation, Visualization, Methodology, Writing - original draft, Project administration; Ruth Styfhals, Conceptualization, Software, Formal analysis, Investigation, Methodology, Writing - review and editing; Ali Murat Elagoz, Writing - review and editing; Gregory E Maes, Resources, Software, Funding acquisition, Writing - review and editing; Eve Seuntjens, Conceptualization, Supervision, Funding acquisition, Project administration, Writing - review and editing

### Author ORCIDs

Astrid Deryckere https://orcid.org/0000-0001-6227-7759
Ruth Styfhals http://orcid.org/0000-0002-5287-8830
Ali Murat Elagoz http://orcid.org/0000-0002-1639-0619
Eve Seuntjens https://orcid.org/0000-0002-0126-461X

### Decision letter and Author response

Decision letter https://doi.org/10.7554/eLife.69161.sa1
Author response https://doi.org/10.7554/eLife.69161.sa2

## Additional files

### Supplementary files

• Supplementary file 1. Number of biological replicates (unless specified otherwise) for immunohistochemistry and in situ hybridization experiments. N/A, not applicable; TR, technical replicate.

• Supplementary file 2. Nucleotide sequence of primers used to amplify gene fragments for ISH probes.

• Supplementary file 3. Nucleotide sequence of probes for colorimetric in situ hybridization.

• Supplementary file 4. Nucleotide sequences of HCR probes.

- Supplementary file 5. Accession numbers of protein sequences used for phylogenetic tree construction.
- Supplementary file 6. Protein sequences of *O. vulgaris* homologs used in phylogenetic tree reconstruction.
- Transparent reporting form

## Data availability

Sequencing data have been deposited in SRA under accession code PRJNA718058.

The following dataset was generated:

| Author(s) | Year | Dataset title | Dataset URL | Database and Identifier |
|---|---|---|---|---|
| Deryckere A, Styfhals R, Elagoz AM, Maes GE, Seuntjens E | 2021 | Identification of neural progenitor cells and their progeny reveals long distance migration in the developing octopus brain (Deryckere et al., 2021) | https://www.ncbi.nlm. nih.gov/bioproject/ 718058 | NCBI BioProject, PRJNA718058 |

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
