## [Decision Letter]

**Acceptance summary:**

This manuscript uses light sheet microscopy, expression patterns of neurogenic genes, and lineage tracing experiments to study the development of the central nervous system in the octopus O. vulgaris. It will be of broad interest to scientists in the field of evolutionary developmental and neurobiology.

**Decision letter after peer review:**

Thank you for submitting your article "Identification of neural progenitor cells and their progeny reveals long distance migration in the developing octopus brain" for consideration by *eLife*. Your article has been reviewed by 3 peer reviewers, including Sonia Sen as the Reviewing Editor and Reviewer #1, and the evaluation has been overseen by Catherine Dulac as the Senior Editor. The following individual involved in review of your submission has agreed to reveal their identity: Shuichi Shigeno (Reviewer #2).

We appreciated the quality, thoroughness and insights that this manuscripts provides. We had a few concerns that the authors could easily address, and these are listed below:

1) What is the nature of the migratory cells? The authors have shown that the migratory cells are largely elav^+^, post-mitotic cells. We're concerned that the low magnification images shown in the figures might miss that there are other cell types that migrate in the transition zones. Could the authors please provide higher magnification images of these regions and comment on whether they think other cell types – for example, glia or progenitors – might also be migrating from the LL to the cords?

2) Is the lateral lip the only neurogenic region? We noticed a low level of expression of ascl1 as well as PCNA in the developing brain cords. Is it possible that there's also a pool of progenitors in the brain cords? Once again, could the authors please provide higher magnification images of this and comment on it? If there are a pool of progenitors that do divide in the developing brain (albeit a smaller number) we suggest modifying the text accordingly.

Related to these essential points are comments in the individual reviews. Could the authors please refer to them while addressing these two main concerns?

*Reviewer #1 (Recommendations for the authors):*

The lineage tracing experiments are beautiful! I had trouble navigating them, though. I found myself going back and forth between figure 1 and 6 to orient myself. It's possible that this is because I am unfamiliar with the neuroanatomy. But maybe the authors could consider unpacking this section a bit? This could be done by splitting this figure. For example – deal with only the optic lobe at stage IX first, then at stage XII. Next deal with PVL/PV similarly?

I got the impression that there might be temporal pattering within the regions as well? For example, PVL neurons seem to be born only at the early time points. What are the authors' thoughts on this?

Do the authors think that the lateral lip is what also contributes to postembryonic neurogenesis?

*Reviewer #2 (Recommendations for the authors):*

1) Classic histological and embryological studies of cephalopod brains have identified a stream-like cell clusters from neurogenic territories surrounding eye regions to the brain, but these regions develop to jaxta-neuronal or neurosecretory tissues called the paravertical body, subpedunculate tissues, neurosecretory tissue of vena cava, and anterior chamber organs as described in Young JZ (1970). Actually, many cells of the subpedunculate tissue are largely embedded in the optic lobes, and cells of the paravertical body are attached to the vertical lobe, and seen in the subvertical and basal lobes. Vascular associated tissues are in the subesophageal mass. Topographically, this study's new terms are corresponding to those developing tissues: lateral lip: LL = sub-buccal tissue and anterior chamber organ, ATZ = paravertical body, and PTX = subpedunculate tissues.

The readers would be convinced that the ectodermal neurogenic regions certainly produced migratory cells for many brain lobes with a stepwise developmental manner as in Figure 6 and 7, but might not be convinced that there are no other cell types migrating from the lateral lip to the brain lobes. Confirming cell types of migratory cells would be essential to propose the main conclusion of this study.

Also, these eye region wrapping optic lobe produce the white body, hematopoietic organs that might be easily labeled with leaked dye and the labeled cells distribute to many brain lobes. The neurotracing (Figure 6) is a really good study and Figure 6I would be the important figure, showing migratory cells may be neurons in the vertical lobe, but the marked cells look big size and not sure these are neurons or endothelial cells of the vasculature. At least a few enlarged figures would be required to characterize the cell types.

2) During the late embryonic stages, the octopus brain continues to develop dramatically to large size, but the expression of neurogenic genes examined in this study disappears. The lateral lip is also very restricted to small areas and the migratory cells can not expect to contribute to major neurogenesis of the brain. One idea may be that weak NeuroD expression and Sox-B1 marked stem-like populations or novel genes are used. This point may be included in the discussion shortly.

In mammalian brain, the migratory cell stream is important for producing certain cell types, e.g. GABAergic neurons in the developing cortex from the ganglionic eminence, but it is not the main contributor to shape the cortex. The ventricular zones mainly produce various cell types as many readers know. The neurogenin and other genes keep being expressed there. However, the octopus developing brain looks very different on this point, indicating octopus neurogenesis still remains a mystery. This view would be helpful for future studies.

In addition to this, there are three points that may increase the impact and justification for this study.

1) In the introductory or late figures, the possible boundaries of LL, ATZ, and PTX could be clearly marked. This is because these are key areas for the main story.

2) I think, as a supplementary, the neurosecretory tissues associated with eye regions could be explained to avoid confusion about what tissues are tracing, although a sentence only for anterior chamber organ is seen in Materials and methods.

3) "The lateral lips harbor the neurogenic zone whereas the developing cords are postmitotic". I almost agree with this, but this statement is a hypothesis where there are neurogenic cells for brain cords. Elav in situ hybridization might not be able to detect a small population of neurogenic cells in the developing brain. In other sentences, the authors need to be careful about progenitor cells that may be in the overall area of the developing brains. Actually an anonymous has a result of BrdU pulse-labeled and positive cells that are distributed in many areas of developing octopus brains, although there is one publication that might show possibly false-positive cells in the vertical lobe. Therefore, neurogenesis for octopus lobes remains controversial and we need more convincing experiments and conclusions.

*Reviewer #3 (Recommendations for the authors):*

Based on anti PH3 and Ov-pcna expression, cell divisions, which might be correlated with neural progenitor formation/division, can be seen across the areas where the supra- and subesophageal ganglia form, and are thus not restricted to the lateral lips. This expression pattern is also acknowledged in the Results section: "While at Stages VII.2 and IX, transcripts were found in most embryonic tissues (mouth apparatus, retina, lateral lips and cords, in the latter at lower intensity), transcripts gradually disappeared in the optic, cerebral, palliovisceral and pedal cords from Stage XI onwards." The absence of cell divisions in later stages does not prove that the neuroepithelium from which the brain potentially develops is not neurogenic, i.e. that all progenitors arise from the lateral lips – they might simply have ceased dividing. Secondly, the expression of genes (SoxB, ASC, neurogenin) that are consistently expressed in neural progenitors across the animal kingdom, are expressed in all areas of the developing brain. Contrary to the description given in the Results section, both Ov-ascl1 and Ov-ngn seem to be expressed in the whole developing brain in stages XV, Ov-ascl1 also in stage XVII and later (Figure 4).

I agree with the authors that the lateral lips are neurogenic regions. The cell lineage tracing data also clearly show the migration of neural precursors into various brain areas (confirming the results by Koenig et al); however, it is also possible that neural progenitors do arise from other regions of the developing brain. This is my main concern – that the data do not convincingly show that the lateral lips are the only areas with neurogenic potential. Corresponding statements/discussions in the results and Discussion sections should be amended accordingly. Additional studies such as cell lineage tracing in the developing cords or laser dissections of the lateral lips would reveal the degree of involvement of the lateral lips in brain neurogenesis.

In this next part of the review, I have subdivided my comments according to the sections in the manuscript.

Abstract

"Our results indicate that progenitors are located outside the central brain cords in the lateral lips adjacent to the eyes, suggesting that newly formed neurons migrate into the cords."

The starting point of the study has to be clarified. It seems that this is not a novel finding (see Koenig et al., 2016)

Introduction

The introduction tries to give an overview of the mechanisms of neurogenesis across the animal kingdom. This is difficult to achieve in a short paragraph given the complexity and diversity of the neural developmental processes and leads to statements that are rather inconclusive, e.g. 'In general, neural progenitor cells are generated from ectodermal cells and divide symmetrically and asymmetrically to generate all neurons of the nervous system'. To improve the introduction, I suggest to limit the background information to lophotrochozoan/molluscan neurogenesis, explain their phylogenetic position and leave the topic of the diversity of neural developmental processes for the discussion.

The authors should also include here that cell lineage tracing in the squid Dorytheutis pealeii has revealed that cells surrounding the eye placode contribute to the anterior chamber organ, the supraesophagial mass, the buccal mass and the buccal ganglion (Koenig et al., 2016).

"Studying species from this group thus brings an opportunity to understand the genetic drivers of the development of organ systems that evolved convergently with vertebrates."

Inaccurate and too vague. Depends on which organ system you are considering and which characters can be traced back to the last common ancestor.

Results

Paragraph 1, lines 91-119 – Please clarify which of the findings are novel. Based on the citations, it seems that the authors are merely confirming previous descriptions of the developing octopus nervous system. If this is the case, the description should be moved into the introduction section. If, on the other hand, the re-examination has yielded new insights, these should be stated clearly.

"Staining intensity increased by Stage XI, showing low-level staining in the lateral lips surrounding the eye placode and more intense staining in all cords and surrounding the mouth. A similar pattern was observed at subsequent stages with highest staining intensity in the central brain cords/masses and low intensity staining in the lateral lips"

Has the staining been quantified? Or is the intensity estimated? Could the difference in intensity be due to the cellular arrangement within the tissue, i.e. areas where more cells accumulate stain more strongly?

"In the central brain, (neuroD) transcripts were present at low level in the optic lobes, supra- and subesophageal masses. A similar expression pattern was visible at Stages XIX.1 and XX.2 with low level expression in all brain masses and clear presence in the transition zones."

The level of expression in the developing brain is the same as for ASC and neurogenin in Stage XV.2 and ASC in Stage XVII but the authors describe the expression of these two genes as absent. (e.g. "At subsequent stages of organogenesis (XI, XIII, XV.2 and XVII), Ov-ascl1 was expressed in the lateral lips and was absent from the developing cords/brain lobes.") This is an important point in the argumentation. Please clarify.

The data on the mitotic activity of subpopulations of neural progenitors expressing asc and neurogenin, respectively, are not convincing. The yellow anti-PH3 staining is so bright that it is not possible to see if it overlaps with the magenta/green of asc/neurogenin. In addition, the statement "Furthermore, co-staining with the PH3 antibody demonstrated that Ov-ascl1^+^ progenitor cells seem more proliferative compared to Ov-ngn^+^ progenitors, that rarely overlap with the PH3" needs additional support. First of all, Ovascl1^+^ progenitors might only divide once but there are simply more of them in the lateral lip. Secondly, the authors would have to analyse different time points since the Ov-ascl1^+^ and the Ov-nngn^+^ populations might be mitotically active at different times.

The corresponding sentence in the Discussion section "In O. vulgaris, the Ov ascl1 progenitor population seems to be more proliferative compared to the Ov-ngn progenitor population, which suggests their sequential expression could be turned around in cephalopods" should be amended accordingly.

"Occasionally, we identified few, single randomly dispersed cells in the pedal or palliovisceral lobes originating from more ventrally located progenitor populations. Considering the very low number of cells compared to the major output from those progenitors, these cells were not taken into account as they might have been labeled while migrating through another area."

If this is the case, the labelling technique needs to be explained better. If progenitors are labeled in a certain area of the lateral lip and their progeny then traced, how can cells then randomly pick up the dye?

Discussion

"This study showed that grossly, the embryonic octopus brain does not contain dividing progenitor

cells.

This is not what the data show and contradicts the statement in the Results section “While at Stages VII.2 and IX, transcripts were found in most embryonic tissues (mouth apparatus, retina, lateral lips and cords, in the latter at lower intensity), transcripts gradually disappeared in the optic, cerebral, palliovisceral and pedal cords from Stage XI onwards."

We further delineated embryonic neurogenesis in O. vulgaris using neural progenitor, pan-neuronal and differentiated-neuron marker genes with conserved functions in neurogenesis, neural specification and differentiation in protostomes and deuterostomes.

Inaccurate. 'Function' should be replaced by 'expression'. Different functions in neurogenesis have been reported for neural genes in different phyla/species and even within the same organism.

The authors also discuss a delay in neural maturation compared to other cephalopods based on the temporal expression of postmitotic neuronal markers; however, without explaining how embryonic stages can be compared between species, specifically in relation to other developmental processes, the discussion is not convincing.

Figures

Figures should be mentioned and sorted in the correct order, including figure panels. Please also consistently include the figure panel you are referring to.

Figure 1

In the transverse sections, the line indicating the subesophageal mass points to a cell-free area. Please correct.

What is the orientation of the reconstruction in panel E? The different lobes are not mentioned in the main text relating to figure 1. Please amend.

---

## [Author Response]

We had a few concerns that the authors could easily address, and these are listed below:1) What is the nature of the migratory cells? The authors have shown that the migratory cells are largely elav^+^, post-mitotic cells. We're concerned that the low magnification images shown in the figures might miss that there are other cell types that migrate in the transition zones. Could the authors please provide higher magnification images of these regions and comment on whether they think other cell types – for example, glia or progenitors – might also be migrating from the LL to the cords?

To clarify this issue, we have performed an additional experiment (including higher magnification images) showing that the lateral lips are neurogenic and that most of the migrating cells are differentiating neurons. Specifically, embryos were fixed 48 hrs after progenitor labeling with CFDA-SE and thin sections were made. We then performed HCR to visualize the expression of Ov-neuroD and Ov-elav and show that the labeled cells in the PTZ indeed express Ov-neuroD and/or Ov-elav (Figure 8 – supplementary figure 2). The CFSE mark revealed that these cells also often presented a neurite-like process. Although we cannot exclude that other cell types are migrating through the PTZ (markers for glia or endothelial cells (cfr Reviewer 2) are not yet known for this species), our data seem to confirm that the bulk of migrating cells in our experiments are neurons. (see also Reviewer 2, question 1 and Reviewer 3, questions 1 and 2).

2) Is the lateral lip the only neurogenic region? We noticed a low level of expression of ascl1 as well as PCNA in the developing brain cords. Is it possible that there's also a pool of progenitors in the brain cords? Once again, could the authors please provide higher magnification images of this and comment on it? If there are a pool of progenitors that do divide in the developing brain (albeit a smaller number) we suggest modifying the text accordingly.

We agree that in the earliest stages (before Stage XI), a limited number of dividing cells is present in the developing cords themselves, and have adapted the text accordingly. From Stage XI onwards, however, we could no longer observe PH3, Ov-pcna, Ov-ascl1, nor Ov-ngn in the cords. As this was a striking observation for us as well, we have confirmed this finding with multiple biological and technical replicates, as well as with a second expression study method (HCR). Taking these data together, we concluded that the low-level staining observed for Ov-pcna, Ov-ascl1 and Ov-ngn in the colorimetric assay is background staining. (see also Reviewer 2, question 5 and Reviewer 3, questions 8 and 9).

Related to these essential points are comments in the individual reviews. Could the authors please refer to them while addressing these two main concerns?Reviewer #1 (Recommendations for the authors):The lineage tracing experiments are beautiful! I had trouble navigating them, though. I found myself going back and forth between figure 1 and 6 to orient myself. It's possible that this is because I am unfamiliar with the neuroanatomy. But maybe the authors could consider unpacking this section a bit? This could be done by splitting this figure. For example – deal with only the optic lobe at stage IX first, then at stage XII. Next deal with PVL/PV similarly?

Thank you for the suggestion to improve the interpretation of the lineage tracing data. We have split the original Figure 6 in two parts, addressing the SEM/SUB (Figure 6) and OL/PdC (Figure 7) separately. In addition, to facilitate navigation through the different lobes in the SEM/SUB, we have reshuffled panels E-F from Figure 1 to Figure 6.

I got the impression that there might be temporal pattering within the regions as well? For example, PVL neurons seem to be born only at the early time points. What are the authors' thoughts on this?

We thank the reviewer for this observation. We indeed have the impression that there is temporal patterning of the progenitors as well. This is the clearest for the optic lobe, where progenitor cells on the posterior side produce cells for the medulla in late stages only. This observation is mentioned in the discussion (line 543-544) and added to the introduction (line 89) and results (line 354) sections. For the palliovisceral lobe, this phenomenon is still preliminary. It indeed looks like cells destined to this lobe are produced at early stages only, but as the progenitor region seems small, more experiments are required to confirm this observation.

Do the authors think that the lateral lip is what also contributes to postembryonic neurogenesis?

As the lateral lips shrink to almost completely disappear at hatching, we suspect that there must be a different, new source of progenitor cells that contributes to neurogenesis during the paralarval and adult phases. Current literature describes dividing cells in the adult brain lobes, but we could not replicate these results and question their validity (see also Reviewer 2, question 5). Although we believe that identifying the source of postembryonic neurons is an important question to solve, the focus of this paper is to characterize embryonic neurogenesis. We have added a supporting sentence at the end of the discussion (line 548-550).

Reviewer #2 (Recommendations for the authors):1) Classic histological and embryological studies of cephalopod brains have identified a stream-like cell clusters from neurogenic territories surrounding eye regions to the brain, but these regions develop to jaxta-neuronal or neurosecretory tissues called the paravertical body, subpedunculate tissues, neurosecretory tissue of vena cava, and anterior chamber organs as described in Young JZ (1970). Actually, many cells of the subpedunculate tissue are largely embedded in the optic lobes, and cells of the paravertical body are attached to the vertical lobe, and seen in the subvertical and basal lobes. Vascular associated tissues are in the subesophageal mass. Topographically, this study's new terms are corresponding to those developing tissues: lateral lip: LL = sub-buccal tissue and anterior chamber organ, ATZ = paravertical body, and PTX = subpedunculate tissues.The readers would be convinced that the ectodermal neurogenic regions certainly produced migratory cells for many brain lobes with a stepwise developmental manner as in Figure 6 and 7, but might not be convinced that there are no other cell types migrating from the lateral lip to the brain lobes. Confirming cell types of migratory cells would be essential to propose the main conclusion of this study.Also, these eye region wrapping optic lobe produce the white body, hematopoietic organs that might be easily labeled with leaked dye and the labeled cells distribute to many brain lobes. The neurotracing (Figure 6) is a really good study and Figure 6I would be the important figure, showing migratory cells may be neurons in the vertical lobe, but the marked cells look big size and not sure these are neurons or endothelial cells of the vasculature. At least a few enlarged figures would be required to characterize the cell types.

We thank the reviewer for this detailed concern. To our understanding, the embryonic origin of those adult juxta-neuronal and neurosecretory tissues has not been experimentally traced. Our study could only trace cells until Stage XIX.2 (right before hatching), and thus cannot provide the evidence that these adult tissues are indeed of lateral lip origin. Concerning the morphology of cells: we observed neurite-like extensions in the migrating cells. However, this observation comes with some caveats: in embryonic stages, morphology of cells might be different from the much better characterized adult cell morphology of these different cell types. It is therefore difficult, or at least insufficient, to use morphology as the sole proxy for determining cell fate. Cells in the lateral lips are packed and have large nuclei and small cytoplasmic areas. Expression of neural markers is extensive. We therefore suspect that the bulk of the cells generated in or migrating out of the lateral lips will be of neural fate, and at least for those periods that we traced the cells, we showed experimental evidence. Indeed, to prove that the lateral lips are neurogenic and that the migrating cells are differentiating neurons, we performed an additional tracing and HCR experiment (Figure 8 – supplementary figure 2). Embryos were fixed 48 hrs after progenitor labeling and thin sections were made. We then performed HCR to visualize the expression of Ov-neuroD and Ov-elav. Although this experiment cannot exclude that other cell types are produced in the lateral lips, we clearly show that the CFSE labeled cells in the PTZ do express Ov-neuroD and/or Ov-elav (line 367-371).

Future identification of markers of glial or endothelial cells should enable us to document the origin of those cells and their contribution to embryonic brain development.

2) During the late embryonic stages, the octopus brain continues to develop dramatically to large size, but the expression of neurogenic genes examined in this study disappears. The lateral lip is also very restricted to small areas and the migratory cells can not expect to contribute to major neurogenesis of the brain. One idea may be that weak NeuroD expression and Sox-B1 marked stem-like populations or novel genes are used. This point may be included in the discussion shortly.In mammalian brain, the migratory cell stream is important for producing certain cell types, e.g. GABAergic neurons in the developing cortex from the ganglionic eminence, but it is not the main contributor to shape the cortex. The ventricular zones mainly produce various cell types as many readers know. The neurogenin and other genes keep being expressed there. However, the octopus developing brain looks very different on this point, indicating octopus neurogenesis still remains a mystery. This view would be helpful for future studies.

We thank the reviewer for raising this concern and suggestion. We believe that most larval brain neurons are produced by the late embryonic stages. Next to a limited number of neurons added by the last progenitors in the lateral lips during maturation stages, the increase in brain size could also result from an increase in cell size, and increased connectivity (see Figure 2, showing a tremendous growth of neuropil stained by an AcTub antibody).

The lateral lips do decrease in size until they disappear almost completely from the hatchling, suggesting that embryonic and post-hatching neurogenesis use a different stem cell pool. In our opinion, neurogenesis post-hatching is most likely re-initiated, from a hitherto unknown population of cells (see also our response to comment 3 below). Further investigation of this phenomenon is out of the scope of this paper. We have added a supporting sentence at the end of the discussion (line 548-550).

The widespread expression of SoxB1 in the brain suggests that soxB1 might participate in the differentiation process of neurons. A similar, dual role for soxB1 (neural stem cell/ differentiation of neurons) has been described for other invertebrate species as well (cfr discussion lines 424-432). Whether these cells are then reactivating the cell cycle later on remains to be determined.

In addition to this, there are three points that may increase the impact and justification for this study.1) In the introductory or late figures, the possible boundaries of LL, ATZ, and PTX could be clearly marked. This is because these are key areas for the main story.

The discovery of the boundaries of these domains happens throughout the story in this paper and the only way to reliably indicate the boundaries of LL, ATZ and PTZ is through in situ hybridization with Ov-ascl1 and Ov-elav or Ov-neuroD. To introduce the naming, we have therefore opted to first add these three different domains to the schematic illustration in Figure 1C, and kept general annotations for the subsequent figures. In Figures 5 and 8, after the molecular boundaries are set, the LL and PTZ boundaries are clearly indicated.

2) I think, as a supplementary, the neurosecretory tissues associated with eye regions could be explained to avoid confusion about what tissues are tracing, although a sentence only for anterior chamber organ is seen in Materials and methods.

To our understanding, the embryonic origin of the adult juxta-neuronal and neurosecretory tissues that the reviewer is referring to, has not been experimentally traced. We can therefore not make an argument that they arise from the lateral lips. We have specifically explained this for the anterior chamber organ/lateral lips in the Results section (line 110-121).

3) "The lateral lips harbor the neurogenic zone whereas the developing cords are postmitotic". I almost agree with this, but this statement is a hypothesis where there are neurogenic cells for brain cords. Elav in situ hybridization might not be able to detect a small population of neurogenic cells in the developing brain. In other sentences, the authors need to be careful about progenitor cells that may be in the overall area of the developing brains. Actually an anonymous has a result of BrdU pulse-labeled and positive cells that are distributed in many areas of developing octopus brains, although there is one publication that might show possibly false-positive cells in the vertical lobe. Therefore, neurogenesis for octopus lobes remains controversial and we need more convincing experiments and conclusions.

Our data show a limited number of proliferating cells at very early stages (before Stage XI). We believe these cells build the initial cord structures as a scaffold upon which the bulk of neurons generated later and arriving from the lateral lips are actually forming the brain. Nevertheless, we agree that we cannot exclude a very small population of neurogenic cells in the cords. Therefore, we changed the subtitle to “The lateral lips harbor proliferating cells whereas the developing cords are mostly postmitotic”. (line 168). In our opinion, the potential contribution of a cordal neurogenic population would be very low and by far outcompeted by the influx of neurons from the lateral lips since dividing cells in the developing cords are almost absent from Stage XI onwards, after which the brain grows tremendously.

We are also convinced that Ov-ascl1, nor Ov-ngn is expressed in the developing brain, already from Stage IX onwards, with the low signal observed in the cords being background (by comparing multiple rounds of colorimetric ISH, see lines 259, 268). Although we cannot exclude low level expression in the brain, our observation is confirmed by independent HCR experiments, in which we did not identify significant Ov-ascl1 and Ov-ngn expression in the brain either (e.g. Figure 5).

In the framework of another study, we have tried to reproduce the adult BrdU and PCNA published data (Bertapelle et al. 2017 and Di Cosmo et al., 2018), but could not find any evidence that these are reproducible. Specifically, using the same PCNA antibody and protocol as published, we did not identify PCNA positive cells in multiple adult brains. Additionally, our Ov-pcna probe for in situ hybridization that shows very strong signal on our embryonic sections did not reveal pcna expression in adult brain slices. Lastly, we have performed numerous experiments with BrdU or EdU on Octopus vulgaris embryos, but so far, these have not resulted in labeled cells. We thus agree with the reviewer that those published data might be false-positive and require additional, more convincing experiments.

As mentioned before, further investigation of this neurogenesis beyond hatching is out of the scope of this paper. We have added a supporting sentence at the end of the discussion (line 548-550).

Reviewer #3 (Recommendations for the authors):Based on anti PH3 and Ov-pcna expression, cell divisions, which might be correlated with neural progenitor formation/division, can be seen across the areas where the supra- and subesophageal ganglia form, and are thus not restricted to the lateral lips. This expression pattern is also acknowledged in the Results section: "While at Stages VII.2 and IX, transcripts were found in most embryonic tissues (mouth apparatus, retina, lateral lips and cords, in the latter at lower intensity), transcripts gradually disappeared in the optic, cerebral, palliovisceral and pedal cords from Stage XI onwards." The absence of cell divisions in later stages does not prove that the neuroepithelium from which the brain potentially develops is not neurogenic, i.e. that all progenitors arise from the lateral lips – they might simply have ceased dividing. Secondly, the expression of genes (SoxB, ASC, neurogenin) that are consistently expressed in neural progenitors across the animal kingdom, are expressed in all areas of the developing brain. Contrary to the description given in the Results section, both Ov-ascl1 and Ov-ngn seem to be expressed in the whole developing brain in stages XV, Ov-ascl1 also in stage XVII and later (Figure 4).I agree with the authors that the lateral lips are neurogenic regions. The cell lineage tracing data also clearly show the migration of neural precursors into various brain areas (confirming the results by Koenig et al); however, it is also possible that neural progenitors do arise from other regions of the developing brain. This is my main concern – that the data do not convincingly show that the lateral lips are the only areas with neurogenic potential. Corresponding statements/discussions in the results and Discussion sections should be amended accordingly. Additional studies such as cell lineage tracing in the developing cords or laser dissections of the lateral lips would reveal the degree of involvement of the lateral lips in brain neurogenesis.

We agree with the reviewer that other structures in the head region bear proliferating cells and that at very early stages, a limited number of proliferating cells can be found in the developing brain. Therefore, we changed the subtitle to “The lateral lips harbor proliferating cells whereas the developing cords are mostly postmitotic”. (line 168), and have made several adjustments to the text (line 22-23 in the abstract, line 172 and 200 in the Results section and line 489-490 in the discussion). However, dividing cells are almost absent from Stage XI onwards, after which the brain grows tremendously. It is possible that these cells cease to divide, but they do not seem to re-enter the cell cycle during embryonic development and can therefore not account for the majority of neurons in the hatchling brain. In addition, there are indeed other regions in the head that contain dividing cells. The mouth region stands out and we have performed preliminary lineage tracing experiments from that region and found that progeny of these cells mostly integrates in the gastrointestinal tract.

We are also convinced that Ov-ascl1, nor Ov-ngn is expressed in the developing brain, already from Stage IX, with the low signal observed in the cords being background (by comparing multiple rounds of ISH, see lines 259, 268). Although we cannot exclude low level expression in the brain, our observation is confirmed by independent HCR experiments, in which we did not identify significant Ov-ascl1 and Ov-ngn expression in the brain either (e.g. Figure 5).

We agree that laser dissections of (parts of) the lateral lips would be a great idea to investigate the contribution of the lateral lips in more detail. For the moment, these experiments remain technically extremely challenging as embryos do not develop well outside of their chorion.

In this next part of the review, I have subdivided my comments according to the sections in the manuscript.Abstract"Our results indicate that progenitors are located outside the central brain cords in the lateral lips adjacent to the eyes, suggesting that newly formed neurons migrate into the cords."The starting point of the study has to be clarified. It seems that this is not a novel finding (see Koenig et al., 2016)

We agree with the reviewer that the starting point of the study could be better clarified. However, since Koenig et al. did not explicitly show the presence of progenitor cells in the lateral lips (with molecular expression data for example), but merely the output of cells from the lateral lips, we have adjusted line 24-26 in the abstract instead, to clarify that our observations align with previous findings in the squid.

IntroductionThe introduction tries to give an overview of the mechanisms of neurogenesis across the animal kingdom. This is difficult to achieve in a short paragraph given the complexity and diversity of the neural developmental processes and leads to statements that are rather inconclusive, e.g. 'In general, neural progenitor cells are generated from ectodermal cells and divide symmetrically and asymmetrically to generate all neurons of the nervous system'. To improve the introduction, I suggest to limit the background information to lophotrochozoan/molluscan neurogenesis, explain their phylogenetic position and leave the topic of the diversity of neural developmental processes for the discussion.

We agree that neural development is diverse across the animal kingdom, but are also convinced that a short summary is helpful to understand the content of the paper, even in very general terms. The goal of this paper is to better understand how complex brains develop, using octopus as an invertebrate with a large and complex brain. We therefore believe that comparison with different mollusks alone in the introduction does not really fit with this story’s objective and prefer to keep the general introduction to facilitate data interpretation. We then review and compare mechanisms of neurogenesis in other lophotrochozoans in the discussion.

The authors should also include here that cell lineage tracing in the squid Dorytheutis pealeii has revealed that cells surrounding the eye placode contribute to the anterior chamber organ, the supraesophagial mass, the buccal mass and the buccal ganglion (Koenig et al., 2016).

We have added this information to the introduction (line 79-82)

"Studying species from this group thus brings an opportunity to understand the genetic drivers of the development of organ systems that evolved convergently with vertebrates."Inaccurate and too vague. Depends on which organ system you are considering and which characters can be traced back to the last common ancestor.

We adapted “organ systems” to “nervous systems” (line 32)

ResultsParagraph 1, lines 91-119 – Please clarify which of the findings are novel. Based on the citations, it seems that the authors are merely confirming previous descriptions of the developing octopus nervous system. If this is the case, the description should be moved into the introduction section. If, on the other hand, the re-examination has yielded new insights, these should be stated clearly.

In this section, we have attempted to describe our own histological data in Figure 1 in the context of different cephalopod species, or adults as referred to in the text. The combination of 2D and 3D images is unique and an added value here, and therefore, we are convinced this description fits best in the Results section. To clarify which are new insights, we have made amendments in lines 98-100 and 116-118.

"Staining intensity increased by Stage XI, showing low-level staining in the lateral lips surrounding the eye placode and more intense staining in all cords and surrounding the mouth. A similar pattern was observed at subsequent stages with highest staining intensity in the central brain cords/masses and low intensity staining in the lateral lips"Has the staining been quantified? Or is the intensity estimated? Could the difference in intensity be due to the cellular arrangement within the tissue, i.e. areas where more cells accumulate stain more strongly?

The staining intensity was not quantified, and the description is based on qualitative observations. The colorimetric in situ hybridization experiments are performed using an automated platform (Ventana Discovery, Roche). This setup increases standardization and thus control over the process. Some batch effects are still observed, but we do not observe differences in staining intensities between sections on a single slide and can therefore at least compare staining intensity between different areas within a section.

Regarding cellular arrangement, the brain, lateral lips and transition zones all have high cell density as can be seen in Figure 1A (DAPI stain). The expression of Ov-ascl1 is very strong in the lateral lips, where Ov-elav is low. We therefore think accumulation differences do not account for the observed intensity differences in our samples. Furthermore, we have been able to reproduce these expression patterns in multiple biological replicates, reinforcing the validity of our observations. Signal that did not consistently appear in all replicates, was considered as background (see also our next response).

Since we did not quantify staining intensity, we adapted the sentence to “strong staining was observed from Stage XI onwards, with …” (line 140).

"In the central brain, (neuroD) transcripts were present at low level in the optic lobes, supra- and subesophageal masses. A similar expression pattern was visible at Stages XIX.1 and XX.2 with low level expression in all brain masses and clear presence in the transition zones."The level of expression in the developing brain is the same as for ASC and neurogenin in Stage XV.2 and ASC in Stage XVII but the authors describe the expression of these two genes as absent. (e.g. "At subsequent stages of organogenesis (XI, XIII, XV.2 and XVII), Ov-ascl1 was expressed in the lateral lips and was absent from the developing cords/brain lobes.") This is an important point in the argumentation. Please clarify.

Comparing ISH performed on replicate samples, it was clear that the Ov-ascl1 and Ov-ngn signal observed in the brain in some batches is background signal (lines 262,271), while for Ov-neuroD, there is specific signal on the edges of each brain lobe (line 292-294). Using HCR as an independent method, we confirmed that Ov-ascl1 and Ov-ngn mRNA expression in the central brain is below detection level, reassuring the lower intensity signal in the central brain on the colorimetric ISH is background. Ov-neuroD expression on the other hand is consistently observed at low-level in the central brain using these two different techniques (see also Figure 5).

The data on the mitotic activity of subpopulations of neural progenitors expressing asc and neurogenin, respectively, are not convincing. The yellow anti-PH3 staining is so bright that it is not possible to see if it overlaps with the magenta/green of asc/neurogenin. In addition, the statement "Furthermore, co-staining with the PH3 antibody demonstrated that Ov-ascl1^+^ progenitor cells seem more proliferative compared to Ov-ngn^+^ progenitors, that rarely overlap with the PH3" needs additional support. First of all, Ovascl1^+^ progenitors might only divide once but there are simply more of them in the lateral lip. Secondly, the authors would have to analyse different time points since the Ov-ascl1^+^ and the Ov-nngn^+^ populations might be mitotically active at different times.The corresponding sentence in the Discussion section "In O. vulgaris, the Ov ascl1 progenitor population seems to be more proliferative compared to the Ov-ngn progenitor population, which suggests their sequential expression could be turned around in cephalopods" should be amended accordingly.

We thank the reviewer to point out that these data deserved a more clear depiction. We have made several adjustments to figure 5 to make overlap between PH3 and the marker, or absence thereof, more convincing. First, we chose to show only one optical section instead of a z-projection in order to delineate the cells better. Second, we marked the PH3 signal on ascl1/neurogenin as well as on neuroD and enlarged different panels so that overlap, or absence of overlap, is more clear.

In addition, we amended the text in the results (line 305, 309) and discussion (line 446) sections to clarify that analysis was performed at Stage XV.2, in order to avoid making overstatements.

"Occasionally, we identified few, single randomly dispersed cells in the pedal or palliovisceral lobes originating from more ventrally located progenitor populations. Considering the very low number of cells compared to the major output from those progenitors, these cells were not taken into account as they might have been labeled while migrating through another area."If this is the case, the labelling technique needs to be explained better. If progenitors are labeled in a certain area of the lateral lip and their progeny then traced, how can cells then randomly pick up the dye?

The CFDA-SE dye is not selective for cell types, and will label any cell present at the location where it is introduced (progenitor or not). However, it labels very locally and only acutely, as the dye is immediately cleaved by esterases and binds cytoplasmic proteins locking it inside cells (information added to line 671-675). We could not observe leakiness, confirming it labels only cell types present in a particular lateral lip area at the time of injection. Based on our hypothesis, these are foremost neural progenitor cells, but our data also suggest there is long distance migration in the lateral lips and precise entry areas into the central brain. Therefore, we cannot exclude that we might have labeled few travelling cells passing by the injection site as well. In any case, as we cannot distinguish travellers from locally generated cells, we cannot make the assumption that these few “outlier” cells “might have been labeled while migrating through another area”. We have therefore adjusted this part of the sentence (line 343).

Discussion"This study showed that grossly, the embryonic octopus brain does not contain dividing progenitor cells.This is not what the data show and contradicts the statement in the Results section 'While at Stages VII.2 and IX, transcripts were found in most embryonic tissues (mouth apparatus, retina, lateral lips and cords, in the latter at lower intensity), transcripts gradually disappeared in the optic, cerebral, palliovisceral and pedal cords from Stage XI onwards."

We agree that the word “grossly” could be misleading. We have adapted the sentence to “This study showed that the embryonic octopus brain is practically devoid of dividing cells from Stage XI onwards” to avoid making overstatements. (line 377-378).

We further delineated embryonic neurogenesis in O. vulgaris using neural progenitor, pan-neuronal and differentiated-neuron marker genes with conserved functions in neurogenesis, neural specification and differentiation in protostomes and deuterostomes.Inaccurate. 'Function' should be replaced by 'expression'. Different functions in neurogenesis have been reported for neural genes in different phyla/species and even within the same organism.

We thank the reviewer for pointing this out and have adapted the text accordingly (line 381).

The authors also discuss a delay in neural maturation compared to other cephalopods based on the temporal expression of postmitotic neuronal markers; however, without explaining how embryonic stages can be compared between species, specifically in relation to other developmental processes, the discussion is not convincing.

We strongly agree with the reviewer that comparing different cephalopod species is not straightforward. We therefore performed a diligent analysis, comparing multiple characteristics of developing cuttlefish, squid and octopuses and published this last year (Deryckere et al., 2020, BMC Developmental Biology). This staging scale now allows reliable comparison of developing embryos across cephalopod orders. We now clearly mentioned this in the discussion (line 407-408).

FiguresFigures should be mentioned and sorted in the correct order, including figure panels. Please also consistently include the figure panel you are referring to.

We have annotated all figure panels and referred to each panel accordingly (see blue highlights in the text).

Figure 1In the transverse sections, the line indicating the subesophageal mass points to a cell-free area. Please correct.

The figure was adapted accordingly.

What is the orientation of the reconstruction in panel E? The different lobes are not mentioned in the main text relating to figure 1. Please amend.

Together with a comment from Reviewer 1, we decided to reshuffle Figure 1. Panels E-F are now part of Figure 6, and panel E (now panel B) contains the appropriate axes, and the different lobes are mentioned in the text accordingly (line 320-324).